# Antiproliferative activity of *Syzygium coriaceum*, an endemic plant of Mauritius, with its UPLC-MS metabolite fingerprint: A mechanistic study

**Nawraj Rummun**[1,2], **Ahmed Serag**[3], **Philippe Rondeau**[4], **Srishti Ramsaha**[1,2], **Emmanuel Bourdon**[4], **Theeshan Bahorun**[2], **Mohamed A. Farag**[5,6]*, **Vidushi S. Neergheen**[2]*

**1** Faculty of Medicine and Health Sciences, Department of Health Sciences, University of Mauritius, Réduit, Republic of Mauritius, **2** Biopharmaceutical Unit, Centre for Biomedical and Biomaterials Research, MSIRI Building, University of Mauritius, Réduit, Republic of Mauritius, **3** Faculty of Pharmacy, Analytical Chemistry Department, Al-Azhar University, Cairo, Egypt, **4** Université de La Réunion, INSERM, UMR 1188 Diabète athérothrombose Thérapies Réunion Océan Indien (DéTROI), Saint-Denis de La Réunion, France, **5** Pharmacognosy Department, College of Pharmacy, Cairo University, Kasr el Aini St., P.B., Cairo, Egypt, **6** Chemistry Department, School of Sciences & Engineering, The American University in Cairo, New Cairo, Egypt

* v.neergheen@uom.ac.mu (VSN); mohamed.alifarag@aucegypt.edu (MAF)

**Data Availability Statement:** All relevant data are within the manuscript and its Supporting information files.

## Abstract

Flowering plants from the *Syzygium* genus have long been used in different ethnomedicinal systems worldwide and have been under scrutiny for their biological activities. *Syzygium coriaceum*, an endemic plant of Mauritius has been poorly studied for its potential application against cancer. Herein, *Syzygium coriaceum* leaf extract has been investigated for its anticancer effect against hepatocellular carcinoma (HepG2) cells. The anticancer activity was assessed using cell proliferation assays, flow cytometry, JC-1 mitochondrial membrane potential assay, and the COMET assay. Un-targeted metabolite profiling *via* ultra-performance liquid chromatography coupled to high-resolution qTOF-MS (UPLC-MS) and aided by molecular networking was employed to identify the crude extract metabolites. *S. coriaceum* treatment induced a dose-dependent increase in lactate dehydrogenase leakage into the culture media, peaking up to 47% (p ≤ 0.0001), compared to untreated control. Moreover, at 40 µg/mL, *S. coriaceum* led to 88.1% (p ≤ 0.0001) drop in mitochondrial membrane potential and 5.7% (p ≤ 0.001) increased in the number of the cell population in G0/G1 phase as well as increased (p < 0.05) the proportion of cells undergoing apoptotic/necrotic cell death. More so, at 10 µg/mL, *S. coriaceum* induced DNA damage which was 19 folds (p < 0.001) higher than that of untreated control cells. Metabolite profiling indicated the presence of 65 metabolites, out of which 59 were identified. Tannins, flavonoids, nitrogenous compounds, and organic acids were the most predominant classes of compounds detected. Our findings showed that the presence of tannins and flavonoids in *S. coriaceum* leaf extract could account for the multiple mechanisms of actions underlying the antiproliferative effect against HepG2 cells.

**Funding:** This study was supported by the Mauritius Research and Innovation Council under the National Research and Innovation Chair Program studentship awarded to NR. MAF acknowledges the financial support received from the Alexander von Humboldt Foundation, Germany. The funders had no role in study design, data collection and analysis, decision to publish, or preparation of the manuscript.

**Competing interests:** The authors have declared that no competing interests exist.

## Introduction

Cancer continues to be a major public threat with an incidence of 18.1 million cases and accounting for 1 in 6 deaths globally, for the year 2018 [1]. This situation warrants more research focusing on drug discovery from unexplored plants for either prevention and/or treatment. Fifteen of the plant-derived oncologic drugs approved since 1980 originated from plants with long medicinal history [2]. Moreover, under their health-promoting activities, numerous herbal extracts emerged putatively as nutraceuticals to serve either as adjuvant therapies or in chemoprevention [3].

Mauritius island which is part of the Mascarene archipelago in the Indian Ocean has the largest flora representing 691 species with 273 species strictly endemic to the island and 150 species endemic to the archipelago [4]. This unique flora has well adapted to the tropical climate with considerable environmental stress induced by UV radiation which has led to remarkable levels of diverse secondary metabolites in particular polyphenols [5]. Polyphenols have been long discussed in the management of cancer, due to their ability to target multiple aberrant proteins and enzymes involved in carcinogenesis. In this regard, the anticancer potential of *Syzygium coriaceum* Bosser & J. Guého against HepG2 cells was investigated in relation to its metabolite fingerprint.

*S. coriaceum* belongs to the Myrtaceae family. *Syzygium* Gaertn., which is the largest flowering genus within the family, encompasses about 1200 globally distributed species with 16 species native to Mauritius [6]. Plants from the genus *Syzygium* have particularly a long ethnomedicinal history worldwide with several reported biological activities [7,8]. Extracts from *S. fruticosum* and *S. cumini* exhibited anti-breast cancer activity in rat models, while *S. aromaticum* showed anti-colon cancer activity [8]. Leaf extract of *S. aqueum* has shown analgesic and antinociceptive potential in male Sprague-Dawley rats [9].

Investigation of *Syzygium* species native to Mauritius island revealed leaf extracts from *S. commersonii*, *S. venosum*, *S. mauritianum* and *S. glomeratum* exhibit potent *in-vitro* antioxidant activities in terms of their reducing potential and free radical scavenging activity [10,11]. Likewise, the potent *in-vitro* activity of *S. coriaceum* leaf extract has been reported in multiple antioxidant assay models [5]. These findings further corroborated in a recent comparative study, whereby *S. coriaceum* was significantly more potent, compared to *S. bijouxii* and *S. pyneei*, in an array of six *in-vitro* antioxidant assay models [12]. Moreover, *S. coriaceum* leaf extract has been reported to significantly potentiate the transcriptional activities of antioxidant enzymes, notably glutathione peroxidase, in cultured COS7 cells [13].

Besides the aforementioned ability of *S. coriaceum* to mitigate oxidative stress, the pluripharmacological potential of the plant is also reported. The leaf extract potentiated the antibacterial activity of ampicillin against *Staphylococcus aureus*, *Escherichia coli* and *Pseudomonas aerigunosa* [5]. Furthermore, essential oils isolated from *S. coriaceum* leaf demonstrated potent anti-α-amylase and anti-tyrosinase enzyme activities [14]. The antiproliferative potential of *S. coriaceum* on breast cancer cells revealed that the methanolic leaf extract of *S. coriaceum* decreased the microtubule-associated protein 1 light chain 3, beclin and telomerase gene expressions, as well as inhibiting the antiapoptotic gene expression. The authors attributed the effects to the presence of gallic acid and quercetin [5]. Recently, a preliminary cytotoxic screening of the *S. coriaceum* crude extract and fractions against liposarcoma cells (SW872), lung cancer cells (A549), liver cancer cells (HepG2) and non-malignant human ovarian cells (HOE) showed the extract selectivity against liver cancer cells [12]. The cytotoxic nature of *S. coriaceum* was attributed to its ability to upregulate intracellular oxidative stress level beyond a critical threshold, which was evident by a dose-dependent surge in intracellular ROS level and a parallel decrease in glutathione peroxidase enzyme activity [12].

The phytochemistry of *S. coriaceum* leaf has been reported principally in terms of polyphenolic composition. Two major bioactive components contributing to the cytotoxic activity against HepG2 cells was elucidated as gallic acid and methyl gallate [12]. Several other secondary metabolites have been identified from *S. coriaceum* leaf, notably, gallotannins, quercetin glycosides, kaempferol glycosides, (+)- catechin, (-)- epicatechin gallate, procyanidin B1 dimer, (E)-β- ocimene and α-guaiene amongst others [12–14].

In line with the above background, this work thus aims at assessing the anticancer effects of *S. coriaceum* and its mechanism of action against HepG2 cells related to its bioactive composition. High-resolution UPLC-MS, aided by molecular networking, has been employed to ultimately characterise the bioactive components that could provide a chemical basis of the extract potential anticancer effects. In this context, molecular networking has been implemented, as it is a very efficient tool for the graphical investigation of structurally related metabolites or compound families thus enabling rapid identification of several phytochemicals within complex samples.

## Materials and methods

### Plant material and preparation of extract

Permission to collect endemic plant samples was obtained from the Mauritius National Park Conservation Services, Ministry of Agro-Industry & Food Security, Réduit, Mauritius. Fresh leaves of *S. coriaceum* were collected, in October 2014, at Gaulette serré, near Camp Thorel situated in the district of Moka. A voucher specimen (MAU 0016404) of the plant was deposited and authenticated at the Mauritius herbarium. The fresh leaves were air-dried followed by exhaustive maceration with aqueous methanol (80%, v/v) and freeze-dried as described previously [15]. For cell-based assays, a stock of 20 mg/mL was prepared using water: DMSO (19:1 part) and sterilized using syringe filters (0.2 $\mu$m pore size). For UPLC-MS analysis, 10 mg of extract was dissolved in 1.5 mL 100% HPLC grade methanol containing 10 μg/mL umbelliferone (an internal standard used for relative quantification of UHPLC-MS features) and sonicated for 15 minutes. The extract was centrifuged at 12000 g for 20 minutes followed by filtration through a 22 μm pore size filter (Agilent, USA).

### Identification and characterisation of secondary metabolites *via* UPLC/MS

High-resolution ultra-performance liquid chromatography-mass spectrometry (UPLC-ESI-qTOF-MS) analysis was performed on an Acquity UPLC system (Waters) equipped with an HSS T3 column (100 × 1.0 mm, particle size 1.8 μm; Waters). The analysis was carried out by applying the following binary gradient at a flow rate of 150 μL minutes$^{-1}$: 0–1 minute, isocratic 95 % A (water/formic acid, 99.9/0.1 [v/v]), 5 % B (acetonitrile/formic acid, 99.9/0.1 [v/v]); 1–16 minutes, linear from 5 to 95 % B; 16–18 minutes, isocratic 95 % B; and 18–22 minutes, isocratic 5 % B. The injection volume was 3.1 μL (full loop injection). The system was coupled to a 6540 Agilent Ultra-High-Definition Accurate Mass Q-TOFLC/MS (Palo Alto, CA, USA) equipped with an ESI interface. The operating conditions briefly were: drying nitrogen gas temperature 325˚C with a flow of 10 L/minutes; nebulizer pressure 20 psig; sheath gas temperature 400˚C with a flow of 12 L/ minutes; capillary voltage 4000 V; nozzle voltage 500 V; fragmentor voltage 130 V; skimmer voltage 45 V; octapole radiofrequency voltage 750 V. Data acquisition (2.5 Hz) in profile mode was governed by Masshunter workstation software (Agilent technologies). The spectra were acquired in negative ionization and positive mode, over a mass-to-charge (m/z), range from 50 to 1200. The detection window was set to 100 ppm. For auto-MS/MS analysis, precursor ions were selected in Q1 with an isolation width of ±3–10 Da and fragmented at collision energies of 15–70 eV using argon as the collision gas.

Metabolites were characterized based on their UV/Vis spectra (220–600 nm), mass spectra and MS/MS fragmentation patterns, and spectra of isolated compounds, compared to reference literature and dictionary of natural products database (CRC, Wiley).

## Molecular networking of *S. coriaceum* metabolites

MS/MS raw data files were converted from .D to.mzXML file format using the MSconvert tool in the ProteoWizard project [16]. These files were uploaded to the global natural product social molecular networking (GNPS) online resource (gnps.ucsd.edu) [17] where molecular networks were generated for the positive and negative measurements separately using the following parameters: parent mass tolerance of 0.02 Da and an MS/MS fragment ion tolerance of 0.1 Da to create consensus spectra. A network was then created where edges were filtered to have a cosine score above 0.7 and more than four matched peaks. Further edges between two nodes were kept in the network if and only if each of the nodes appeared in each other's respective top 10 most similar nodes. To visualize the resulted networks, they were imported into Cytoscape software (version 3.7.2) [18] where the nodes correspond to a specific MS/MS spectrum and edges represent the significant pairwise alignment between nodes.

## Cell culture

Human hepatocellular carcinoma cells (HepG2) (ATCC HB-8065) were cultured in Dulbecco's Modified Eagle's medium supplemented with 10 % fetal bovine serum, 2 mM L-glutamine, 100 units/L penicillin and 100 units/L streptomycin. Cells were grown in a humidified atmosphere of 5% carbon dioxide and 95% humidity at 37˚C. For cell treatments, different concentrations of test extracts (10–100 µg/mL, as previously reported [12]), were added to the culture medium for 48 hours, unless otherwise indicated. The negative control comprised of extract vehicle containing 0.125% v/v DMSO, unless otherwise stated. Experiments were carried out in triplicates.

**Lactate dehydrogenase (LDH) assay.** LDH release by HepG2 cells was determined as described [19]. The experimental positive control comprised of cells incubated with 1% Triton X-100. The cytotoxicity in terms of LDH release was expressed as a percentage of the positive control. The LDH activity of the extract was compared to that of etoposide, which is a clinically used chemotherapeutic drug.

**Clonogenic cell survival assay.** The protocol was adapted with slight modifications [20]. HepG2 cells were seeded in a 6-well plate (500/well) and allowed to attach overnight. Following the treatment period of 48 hours with different concentrations of extract (10, 20 and 40 µg/mL, equivalent to ½ X, X and 2X of the reported $IC_{50}$ value) and control, the media was replaced with fresh complete culture media and cells were grown under standard recommended culture conditions for 7 additional days. Colonies were then fixed with 4% paraformaldehyde for 30 minutes and stained with 0.5% (w/v) crystal violet. The individual wells were imaged using a digital camera and the colonies counted using Image J software (the US, National Institute of Health). The cytotoxic effect was expressed as the percentage of surviving colonies relative to control.

**Comet assay.** *Pre-treatment of HepG2 cells.* HepG2 cells (3 x $10^4$ cells/well) were cultured in 12-well plates overnight. Following trial experiments using different dilutions of extract from 10 µg/mL to 40 µg/mL at 24 hours and 48 hours, the cells were exposed to 10 µg/mL extract for 24 hours. The negative control consisted of untreated HepG2 cells while the positive control cells were treated with 200 µM $H_2O_2$ for 30 minutes as previously reported [21]. Cells were washed with PBS, harvested using 1 X trypsin- EDTA and re-suspended in 100 µL of PBS.

## Single cell gel electrophoresis

Comet assay was carried out according to Miyaji et al (2004) [22] with minor modifications. Briefly, 40 μL of a mixture comprising 30 μL of pre-treated cells and 70 μL of 1% (w/v) low melting agarose was placed on a frosted microscope slide, pre-coated with 1.5% normal melting agarose. The mixture was covered with a coverslip and allowed to solidify at 4˚C for 1 hour. Following solidification, the coverslip was slid off and the slides were immersed in pre-chilled lysis buffer (2.5 M NaCl, 0.1 M EDTA, 10 mM Tris base, 1% v/v Triton X-100 (added 30 minutes before use), pH 10.4˚C) for 1 hour. Then, the slides were submerged in pre-chilled electrophoresis buffer (0.2 M NaOH, 1 mM EDTA, pH 13. 4˚C). After 30 minutes of electrophoresis at 30 volts and 350 mA, the slides were immersed in pre-chilled neutralisation buffer (0.4 Tris-HCL, pH 7.5, 4˚C) for 10 minutes, washed in distilled water, fixed with 4% formalin solution for 20 minutes and air-dried overnight. The slides were stained with Hoechst 33342 (1 μg/mL), air-dried and visualised at 200 X magnification in DAPI, using EVOS fluorescence microscope (Life Technologies). Damaged DNA was measured for 100 randomly selected cells (for each independent experiment) using the Comet Assay IV 4.3.1 (Perceptive instrument, UK).

## Flow cytometry analysis

Apoptosis/necrosis analysis was performed on HepG2 cells after 48 hours treatment by flow cytometry (Beckman Coulter's CytoFLEX and Cytexpert software) using Annexin V-FITC and propidium iodide (PI) double staining as described [23]. Cell cycle analysis was performed using propidium iodide for DNA staining. Percentage of cells in different phases ($G_0/G_1$, S and $G_2M$ phases) were quantified from propidium iodide fluorescence intensity-area (PI-A) histograms corresponding to the DNA content of HepG2 cells.

## Mitochondrial membrane potential (MMP) assay

The MMP of HepG2 cells was assessed as described with slight modification [24]. Overnight grown cells (5 x $10^3$ cells/well) in 96-well plate were treated with extracts for 48 hours and stained with 100 μL of 10 μg/mL JC-1 (Invitrogen™; Thermo Fisher Scientific, Inc.) for 30 minutes at 37˚C. The JC-1 fluorescence was measured at wavelengths of 485 nm (excitation)/ 528 nm (emission) (green fluorescence) and 540 nm (excitation)/ 590 nm (emission) (red fluorescence) pairs. The red/green ratios were calculated to indicate the mitochondrial membrane potential.

## Statistical analysis

Statistical analyses were performed using GraphPad Prism 6 software (GraphPad Inc., San Diego, California). The mean values among extract and control were compared using one-way ANOVA. Student t-test and/or Tukey's multiple comparisons as Post Hoc test was used to assess significant differences between the mean value for extract concentrations and that of the negative control.

# Results & discussion

## Metabolite fingerprint of *S. coriaceum* leaf extract

Non-targeted profiling *via* ultra-performance liquid chromatography coupled to high-resolution qTOF-MS was employed to characterise secondary metabolites which could possibly be the elements mediating the antiproliferative activity of *S. coriaceum* leaf crude extract. Hyphenated ultra-performance liquid chromatography (UPLC) with high-resolution mass

spectrometry offers several advantages including high separation efficiency and excellent resolution with relatively short analysis times [25]. High-resolution mass spectrometry warrants resolving of the isomeric and isobaric species due to their distinctive MS/MS fragmentation pattern, a highly informative preference used in metabolites identification [26]. However, manual interpretation of such tandem MS/MS fingerprints is not a trivial task, thus prompting to computer-based approaches to allow the visualization and organization of these data [27].

A gradient mobile phase of acetonitrile with aqueous formic acid followed by (+/−) ESI-qTOF-MS detection allowed for comprehensive detection of *S. coriaceum* leaf metabolites including tannins, flavonoids, amino acids, organic acids and amino alcohols within about 21 minutes. The negative ESI spectra permitted the detection of tannins, organic acids and flavonoids owing to the presence of phenyl and carboxyl groups whereas positive ion mode showed a better performance for amino acids and amino alcohols to be preferentially ionized under these conditions due to the presence of nitrogen atoms [26]. Representative UPLC–MS chromatograms of *S. coriaceum* leaf crude extract detected in both (+/−) ESI modes are presented in Fig 1. Metabolite identification was based on their accurate MS and MS/MS spectra in comparison to our in-house database, phytochemical dictionary of natural products database and reference literature. Moreover, the global natural products social molecular networking (GNPS) was further employed to search tandem MS/MS spectral space of *S. coriaceum* metabolites thus enabling the visual investigation of structurally related metabolites or compound families. A total of 65 chromatographic peaks were annotated, of which 59 were identified with tannins as the most abundant class. The retention time, experimental *m/z*, molecular formula, mass error, main $MS^2$ fragments, relative concentrations and identities for these peaks are

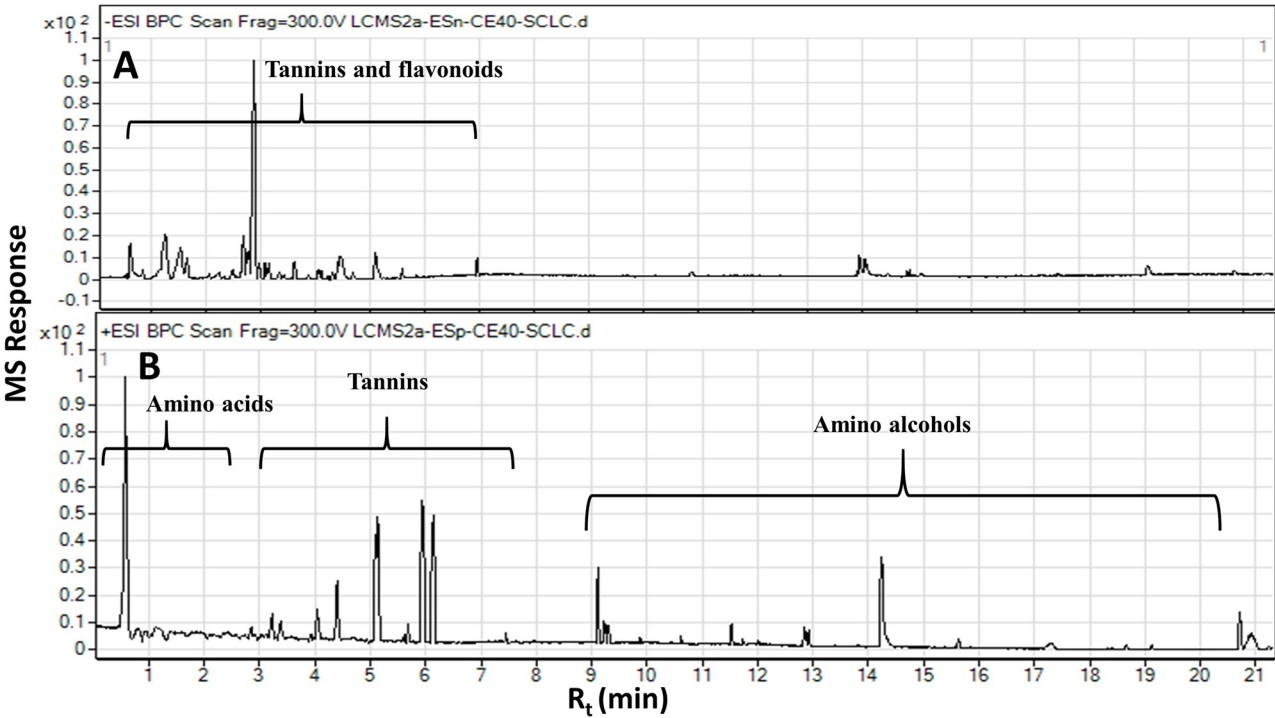

**Fig 1. Representative UPLC-MS base peak traces analyzed in the (A) negative ion mode and (B) positive ion mode of *S. coriaceum* leaf crude extract with peaks mainly due to tannins, flavonoids, amino acids and amino alcohols.**

presented in (Table 1). The basic structures of these metabolites discussed throughout the manuscript are illustrated in Fig 2.

**Tannins.** A total of 29 hydrolysable tannins peaks belonging to two main subclasses *i.e.* ellagitannins and gallotannins were detected in *S. coriaceum* extract. Peaks derived from hexahydroxydiphenoyl hexose (3, 20, 22, 31, 35, 38 and 42), a building block for ellagitannins, have been characterized in this study based on their characteristic MS/MS fingerprints *i.e.* daughter ions at *m/z* 301 and 169 corresponding to hexahydroxydiphenoyl and gallic acid fragments, respectively. Such similarity in the tandem MS/MS pattern has been instrumental to cluster these peaks in a single molecular network group where the nodes of this network to be labelled by compounds' precursor ions and the edges which connect these nodes have been labelled by the mass differences between these compounds as depicted in (Fig 3A). Such graphical display along with the accurate masses and MS/MS fragmentation patterns have resulted in the annotation of theses peaks as follows: mono, di and tri *O*-galloyl-hexahydroxydiphenoyl hexose were assigned for peak 3 (633.0741, $C_{27}H_{21}O_{18}{}^-$), peak 31 (785.0847, $C_{34}H_{25}O_{22}{}^-$) and peak 38 (937.0965, $C_{41}H_{29}O_{26}{}^-$), respectively, with a mass difference of 152 Da indicative for the extra galloyl moieties (S1a–S1c Fig). Moreover, a methoxy derivative of hexahydroxydiphenoyl hexose (peak 20; 815.0953, $C_{35}H_{27}O_{23}{}^-$) was also detected in this cluster assigned as methoxy-di-*O*-galloyl-hexahydroxydiphenoyl hexose with a mass difference of 30 Da from peak 31 indicative for the extra methoxy group (S1d Fig). Additionally, methoxy-tri-*O*-galloyl-hexahydroxydiphenoyl hexose (peak 35; 967.103, $C_{42}H_{31}O_{27}{}^-$) and methoxy-tetra-*O*-galloyl-hexahydroxydiphenoyl hexose (peak 42; 1119.11, $C_{49}H_{35}O_{31}{}^-$) have been also clustered in the same network with a respective mass loses of 152 and 304 Da from peak 20 indicative for the extra galloyl moieties. Interestingly, another two ellagitannins building blocks namely chebulic acid (peak 21; 355.0306, $C_{14}H_{11}O_{11}{}^-$) and ellagic acid (peak 41; 300.9999, $C_{14}H_5O_8{}^-$) have also been detected in the *S. coriaceum* leaf crude extract.

A series of gallotannins precursors such as polygalloyl esters of hexose have been also unveiled based on their MS/MS pattern and assisted by molecular networking (Fig 3B) with, di-*O*-galloyl-hexoside (peak 14; 483.0776, $C_{20}H_{19}O_{14}{}^-$), tri-*O*-galloylhexose (peak 24; 635.0891, $C_{27}H_{23}O_{18}{}^-$), tetra-*O*-galloylhexose (peak 32; 787.0989, $C_{34}H_{27}O_{22}{}^-$) and penta-*O*-galloylhexose (peak 43; 939.1121, $C_{41}H_{31}O_{26}{}^-$) with a mass difference of 152 Da indicative for the extra galloyl moieties in these peaks as previously observed. Also, galloyl-hexoside (peak 9; 331.0669, $C_{13}H_{15}O_{10}{}^-$) has been also annotated based on its MS/MS fragmentation pattern at *m/z* 169 corresponding to the neutral loss of its hexosyl moieties [M-H-162]$^-$ and *m/z* 125 [M-H-162-44]$^-$ (S2a Fig). Similarly, peak 29 displayed [M-H]$^-$ at *m/z* (467.0835, $C_{20}H_{19}O_{13}{}^-$) was assigned as di-*O*-galloyl rhamnoside based on its characteristic fragmentation pattern at *m/z* 169 corresponding to the respective loss of the rhamnosyl and galloyl residues [M-H-146-152]$^-$ (S2b Fig). On the other hand, polygalloyl esters of quinic acid, previously reported in *S. samarangense* [28], and galloyl ester of glycerol were also detected in *S. coriaceum* leaf extract *via* positive ionization mode assigned as tri-*O*-galloylquinic acid (peak 25; 649.1027, $C_{28}H_{25}O_{18}{}^+$) and *O*-galloylglycerol (peak 53; 245.066, $C_{10}H_{13}O_7{}^+$). Gallic acid (peak 13; 169.0143, $C_7H_5O_5{}^-$) and its methylated derivative *i.e.* methyl gallate (peak 26; 183.0309, $C_8H_7O_5{}^-$) have been also characterized with 14 Da mass difference indicative for the extra methyl group. Methyl gallate was the most abundant metabolite quantified followed by gallic acid (Table 1). It is worth noting that high total phenolic content (62 ± 6.04 mg gallic acid equivalent per gram extract) of *S. coriaceum* leaf has been reported; with both gallic acid and methyl gallate identified as major bioactive compounds contributing to its cytotoxicity against HepG2 cells, *via* upregulation of intracellular reactive oxygen species [12]. Likewise, ellagic acid has been shown to suppress the growth of HepG2 cells *via* upregulation of intracellular ROS level, increased caspase- 3/9 activity as well as increased expression of P53, Bax and Cyt-c

**Table 1. Metabolites identified in the crude extract of *S. coriaceum* leaves *via* UPLC- ESI-QToF/MS in negative and positive ionization modes.**

| Peak No. | RT (minutes) | Mol. ion m/z (±) | Molecular formula | Error (ppm) | Name | Class | MS/MS | Relative concentration ± %RSD [a] |
|---|---|---|---|---|---|---|---|---|
| 1 | 0.619 | 195.0519 | $C_6H_{11}O_7^-$ | -4.37 | Gluconic acid | Sugar | 75, 59 | 3.2 ± 2.96 |
| 2 | 0.637 | 191.0572 | $C_7H_{11}O_6^-$ | -5.73 | Quinic acid | Organic acid | 85, 59 | 34.2 ± 12.46 |
| 3 | 0.664 | 633.0741 | $C_{27}H_{21}O_{18}^-$ | -0.85 | Galloyl-*O*-hexahydroxydiphenoyl hexose | Hydrolysable tannin | 301, 275, 249, 231, 169 | 11.6 ± 3.66 |
| 4 | 0.746 | 130.0879 | $C_6H_{12}NO_2^+$ | -13.18 | *N*-methyl proline | Amino acid | 84, 70, 56 | 2.1 ± 51.02 |
| 5 | 0.758 | 136.0629 | $C_4H_{11}NO_4^+$ | -17.45 | Hydroxymethylserine/ Hydroxythreonine. | Nitrogenous compound | 119, 92 | 1.3 ± 10.57 |
| 6 | 0.829 | 173.0456 | $C_7H_9O_5^-$ | -0.12 | Shikimic acid | Organic acid | 93 | 5.2 ± 24.06 |
| 7 | 0.857 | 191.0201 | $C_6H_7O_7^-$ | -1.77 | Citric acid/(iso)Citric acid | Organic acid | 87, 57 | 6.2 ± 3.02 |
| 8 | 0.873 | 203.0196 | $C_7H_7O_7^-$ | 0.45 | Unknown | Unknown | 192, 71 | 1.9 ± 1.6 |
| 9 | 0.879 | 331.0669 | $C_{13}H_{15}O_{10}^-$ | 0.65 | Galloyl-hexoside | Hydrolysable tannin | 169, 125 | 4.8 ± 3.95 |
| 10 | 0.886 | 182.0811 | $C_9H_{12}NO_3^+$ | 1.9 | Tyrosine | Amino acid | 91, 77 | 0.7 ± 4.64 |
| 11 | 0.9 | 268.1036 | $C_9H_{18}NO_8^+$ | -3.46 | Unknown amino sugar | Nitrogenous compound | 136 | 0.3 ± 25.01 |
| 12 | 1.03 | 229.1542 | $C_{11}H_{21}N_2O_3^+$ | 2.0557 | Isoleucylproline | peptide | 70 | 0.7 ± 22.96 |
| 13 | 1.183 | 169.0143 | $C_7H_5O_5^-$ | -0.39 | Gallic acid | Hydrolysable tannin | 124, 108, 78 | 62.2 ± 0.94 |
| 14 | 1.307 | 483.0776 | $C_{20}H_{19}O_{14}^-$ | 0.87 | Di-*O*-galloyl-hexoside | Hydrolysable tannin | 169, 125 | 23.8 ± 1.37 |
| 15 | 1.324 | 207.0499 | $C_7H_{11}O_7^+$ | 0.19 | Homocitric acid | Organic acid | 101, 59 | 0.7 ± 14.3 |
| 16 | 1.436 | 411.0237 | $C_{16}H_{11}O_{13}^-$ | -7.01 | Unknown | Hydrolysable tannin | 241, 169, 125, 97 | 5.4 ± 0.37 |
| 17 | 1.498 | 166.0862 | $C_9H_{12}NO_2^+$ | 0.26 | Phenylalanine | Amino acid | 120, 103, 91, 77 | 2.3 ± 4.79 |
| 18 | 1.617 | 467.0813 | $C_{20}H_{19}O_{13}^+$ | 1.69 | Hexahydroxydiphenoyl-proto-quercitol | Hydrolysable tannin | 153 | 1.4 ± 22.24 |
| 19 | 1.889 | 138.0547 | $C_7H_8NO_2^+$ | 1.71 | Unknown | Nitrogenous compound | 80 | 0.5 ± 3.73 |
| 20 | 2.09 | 815.0953 | $C_{35}H_{27}O_{23}^-$ | 3.01 | Methoxy-di-*O*-galloyl-hexahydroxydiphenoyl hexose | Hydrolysable tannin | 633, 301, 169 | 9.2 ± 0.63 |
| 21 | 2.324 | 355.0306 | $C_{14}H_{11}O_{11}^-$ | 0.18 | Chebulic acid | Hydrolysable tannins | 203, 169, 125 | 4.8 ± 2.52 |
| 22 | 2.529 | 679.1155 | $C_{29}H_{27}O_{19}^-$ | -0.19 | Ethoxygalloyl-*O*-hexahydroxydiphenoyl hexose | Hydrolysable tannin | 301, 275, 169 | 5.6 ± 0.25 |
| 23 | 2.761 | 467.0813 | $C_{20}H_{19}O_{13}^+$ | 1.73 | Hexahydroxydiphenoyl-proto-quercitol isomer | Hydrolysable tannin | 153 | 0.8 ± 8.28 |
| 24 | 2.8 | 635.0891 | $C_{27}H_{23}O_{18}^-$ | 0.03 | Tri-*O*-galloyl-hexoside | Hydrolysable tannin | 313, 169 | 7.9 ± 4.18 |
| 25 | 2.848 | 649.1027 | $C_{28}H_{25}O_{18}^+$ | 1.52 | Tri-*O*-galloylquinic acid | Hydrolysable tannin | 277, 153 | 0.7 ± 13.77 |
| 26 | 2.855 | 183.0309 | $C_8H_7O_5^-$ | -5.23 | Methyl gallate | Hydrolysable tannin | 124, 78 | 150.6 ± 0.97 |
| 27 | 2.868 | 344.1338 | $C_{15}H_{22}NO_8^+$ | 0.91 | *N*-deoxy-fructosyl tyrosine | Acylated amino acid | 147 | 0.9 ± 5.07 |
| 28 | 3.056 | 619.0922 | $C_{27}H_{23}O_{17}^+$ | 1.51 | Galloyl-*O*-hexahydroxydiphenoyl-quercitol | Hydrolysable tannin | 153 | 0.6 ± 14.41 |
| 29 | 3.322 | 467.0835 | $C_{20}H_{19}O_{13}^-$ | -0.48 | Di-*O*-galloyl rhamnoside | Hydrolysable tannin | 169, 152, 125 | 2.9 ± 1.08 |
| 30 | 3.399 | 469.0047 | $C_{21}H_9O_{13}^-$ | 0.5 | Flavogallonic acid | Hydrolysable tannin | 301, 285 | 2.2 ± 2.09 |

*(Continued)*

**Table 1.** (*Continued*)

| Peak No. | RT (minutes) | Mol. ion m/z (±) | Molecular formula | Error (ppm) | Name | Class | MS/MS | Relative concentration ± %RSD [a] |
|---|---|---|---|---|---|---|---|---|
| 31 | 3.432 | 785.0847 | $C_{34}H_{25}O_{22}^{-}$ | -0.15 | Di-*O*-galloyl-hexahydroxydiphenoyl hexose | Hydrolysable tannin | 301, 275, 249, 169 | 7.4 ± 4.62 |
| 32 | 3.674 | 787.0989 | $C_{34}H_{27}O_{22}^{-}$ | 1.25 | Tetra-*O*-galloyl-hexoside | Hydrolysable tannin | 465, 169, 125 | 2.6 ± 2.74 |
| 33 | 3.719 | 282.1181 | $C_{10}H_{20}NO_{8}^{+}$ | 1.09 | Fructosyl-threonine | Amino acid | 113, 85 | 4.1 ± 2.5 |
| 34 | 3.785 | 153.0548 | $C_{8}H_{9}O_{3}^{+}$ | -1.48 | Unknown | Unknown | 113, 85 | 0.6 ± 12.66 |
| 35 | 3.788 | 967.103 | $C_{42}H_{31}O_{27}^{-}$ | -0.6 | Methoxy-tri-*O*-galloyl-hexahydroxydiphenoyl hexose | Hydrolysable tannin | 785, 301 | 4.4 ± 0.51 |
| 36 | 3.929 | 227.1279 | $C_{12}H_{19}O_{4}^{+}$ | -0.4 | Unknown | Unknown | | 1.9 ± 9.17 |
| 37 | 4.013 | 769.0845 | $C_{34}H_{25}O_{21}^{+}$ | 1.7 | Balanophotannin | Hydrolysable tannin | 277, 153 | 0.7 ± 21.44 |
| 38 | 4.052 | 937.0965 | $C_{41}H_{29}O_{26}^{-}$ | -0.85 | Tri-*O*-galloyl-hexahydroxydiphenoyl hexose | Hydrolysable tannin | 785, 301, 275, 169 | 4.5 ± 1.99 |
| 39 | 4.109 | 501.0708 | $C_{23}H_{17}O_{13}^{-}$ | 5.69 | Ethyl-p-trigallate | Hydrolysable tannin | 313, 169 | 1.9 ± 0.57 |
| 40 | 4.512 | 512.9987 | $C_{15}H_{13}O_{20}^{-}$ | 4.23 | Unknown | Unknown | 301 | 6.7 ± 1.88 |
| 41 | 4.565 | 300.9999 | $C_{14}H_{5}O_{8}^{-}$ | -2.69 | Ellagic acid | Hydrolysable tannin | 283, 145 | 36.5 ± 0.91 |
| 42 | 4.69 | 1119.11 | $C_{49}H_{35}O_{31}^{-}$ | -0.42 | Methoxy-tetra-*O*-galloyl-hexahydroxydiphenoyl hexose | Hydrolysable tannin | 937, 749, 579, 301 | 2.0 ± 2.62 |
| 43 | 4.714 | 939.1121 | $C_{41}H_{31}O_{26}^{-}$ | -0.88 | Penta-*O*-galloyl-hexoside | Hydrolysable tannin | 769, 617, 169 | 6.2 ± 1.4 |
| 44 | 4.898 | 653.082 | $C_{23}H_{25}O_{22}^{-}$ | 4.43 | Unknown tannin | Hydrolysable tannin | 169, 125 | 2.1 ± 1.54 |
| 45 | 5.095 | 579.1377 | $C_{26}H_{27}O_{15}^{-}$ | -3.23 | Quercetin-*O*-pentosly-rhamnoside | Flavonoid | 300, 271 | 22.7 ± 0.16 |
| 46 | 5.174 | 447.0941 | $C_{21}H_{19}O_{11}^{-}$ | -1.68 | Quercetin-*O*-rhamnoside | Flavonoid | 300, 271, 255 | 8.7 ± 1.6 |
| 47 | 5.194 | 483.021 | $C_{22}H_{11}O_{13}^{-}$ | -0.75 | Flavogallonic acid methyl ester | Hydrolysable tannin | 301, 270 | 7.5 ± 5.05 |
| 48 | 5.388 | 218.2111 | $C_{12}H_{28}NO_{2}^{+}$ | 1.55 | Amino-dodecanediol | Amino alcohol | 88, 70 | 0.3 ± 1.67 |
| 49 | 5.4 | 573.125 | $C_{27}H_{25}O_{14}^{-}$ | -0.09 | Benzyl-di-*O*-galloyl-hexoside | Hydrolysable tannin | 169, 125 | 1.5 ± 3.45 |
| 50 | 5.597 | 527.0143 | $C_{23}H_{11}O_{15}^{-}$ | -6.65 | Unknown | Unknown | 315, 300 | 7.0 ± 1.76 |
| 51 | 5.625 | 563.1411 | $C_{26}H_{27}O_{14}^{-}$ | -0.63 | Kaempefrol-*O*-pentosyl-rhamnoside | Flavonoid | 284 | 1.6 ± 2.95 |
| 52 | 5.984 | 731.1472 | $C_{33}H_{31}O_{19}^{-}$ | -0.23 | Unknown | Unknown | 579, 300 | 1.4 ± 4.87 |
| 53 | 6.024 | 245.066 | $C_{10}H_{13}O_{7}^{+}$ | -1.65 | *O*-Galloylglycerol | Hydrolysable tannin | 105, 75, 59 | 21.3 ± 9.21 |
| 54 | 6.959 | 469.0813 | $C_{16}H_{21}O_{16}^{-}$ | 5.69 | Unknown | Unknown | 241, 97 | 13.2 ± 3.17 |
| 55 | 7.445 | 246.2425 | $C_{14}H_{32}NO_{2}^{+}$ | 1.16 | Amino-tetradecanediol | Amino alcohol | 88, 70, 57 | 0.4 ± 5.35 |
| 56 | 9.108 | 274.2741 | $C_{16}H_{36}NO_{2}^{+}$ | -0.28 | Amino-hexadecanediol | Amino alcohol | 88, 70, 57 | 7.3 ± 4.57 |
| 57 | 9.162 | 230.2478 | $C_{14}H_{32}NO^{+}$ | 0.44 | Amino-tetradecanol | Amino alcohol | 57 | 0.3 ± 6.99 |
| 58 | 9.239 | 318.3003 | $C_{18}H_{40}NO_{3}^{+}$ | 0.2 | Amino-octadecanetriol | Amino alcohol | 88, 70, 57 | 2.7 ± 6.06 |
| 59 | 9.359 | 362.3261 | $C_{20}H_{44}NO_{4}^{+}$ | 1 | Amino-eicosanetetrol | Amino alcohol | 70, 57 | 0.4 ± 9.5 |
| 60 | 9.61 | 288.2894 | $C_{17}H_{38}NO_{2}^{+}$ | 1.2 | Amino-heptadecanediol | Amino alcohol | 88, 70, 57 | 0.8 ± 41.36 |
| 61 | 10.619 | 302.3053 | $C_{18}H_{40}NO_{2}^{+}$ | 0.36 | Amino-octadecanediol | Amino alcohol | 88, 70, 57 | 1.2 ± 8.34 |
| 62 | 10.718 | 346.3309 | $C_{20}H_{44}NO_{3}^{+}$ | 1.9 | Aminoeicosanetriol | Amino alcohol | 88, 70, 57 | 0.3 ± 10.45 |
| 63 | 12.018 | 330.3365 | $C_{20}H_{44}NO_{2}^{+}$ | 0.58 | Aminoeicosanediol | Amino alcohol | 88, 70, 57 | 1.1 ± 9.73 |
| 64 | 13.392 | 358.3677 | $C_{22}H_{48}NO_{2}^{+}$ | 1.04 | Aminodocosanediol | Amino alcohol | 88, 70, 57 | 0.7 ± 9.21 |

(*Continued*)

**Table 1.** (Continued)

| Peak No. | RT (minutes) | Mol. ion m/z (±) | Molecular formula | Error (ppm) | Name | Class | MS/MS | Relative concentration ± %RSD [a] |
|---|---|---|---|---|---|---|---|---|
| 65 | 21.278 | 338.341 | $C_{22}H_{44}NO^+$ | -0.19 | Docosenamide | Nitrogenous compound | 69, 65 | 0.7 ± 33.49 |

[a] Concentrations of the detected metabolites are calculated based on the peak areas of their electron-ion chromatograms relative to the internal standard (10 µg/mL umbelliferone). %RSD is calculated for three biological replicates.

apoptosis-related proteins, subsequently inducing G0/G1 cell cycle arrest and ultimately apoptotic cell death [29]. However, the contribution of other identified constituents (Table 1), notably, quinic acid, *O*-galloylglycerol, di-*O*-gallolyl-hexoside and quercetin-*O*-rhamnoside cannot be neglected.

**Flavonoids.** In contrast to the abundance of flavonoids in *Syzygium* species *i.e.*, *S. aqueum*, *S. paniculatum*, *S. samarangense*, *S. gratum*, *S. jambos*, *and S. malaccense* [30], very few flavonoid peaks have been annotated in this study suggesting that *S. coriaceum* is not flavonoid rich. All detected flavonoids were of *O*-flavonol type glycosides with characteristic MS/ MS fragmentation pattern of the *O*-glycosidic bond leading to neutral loss of the sugar residues

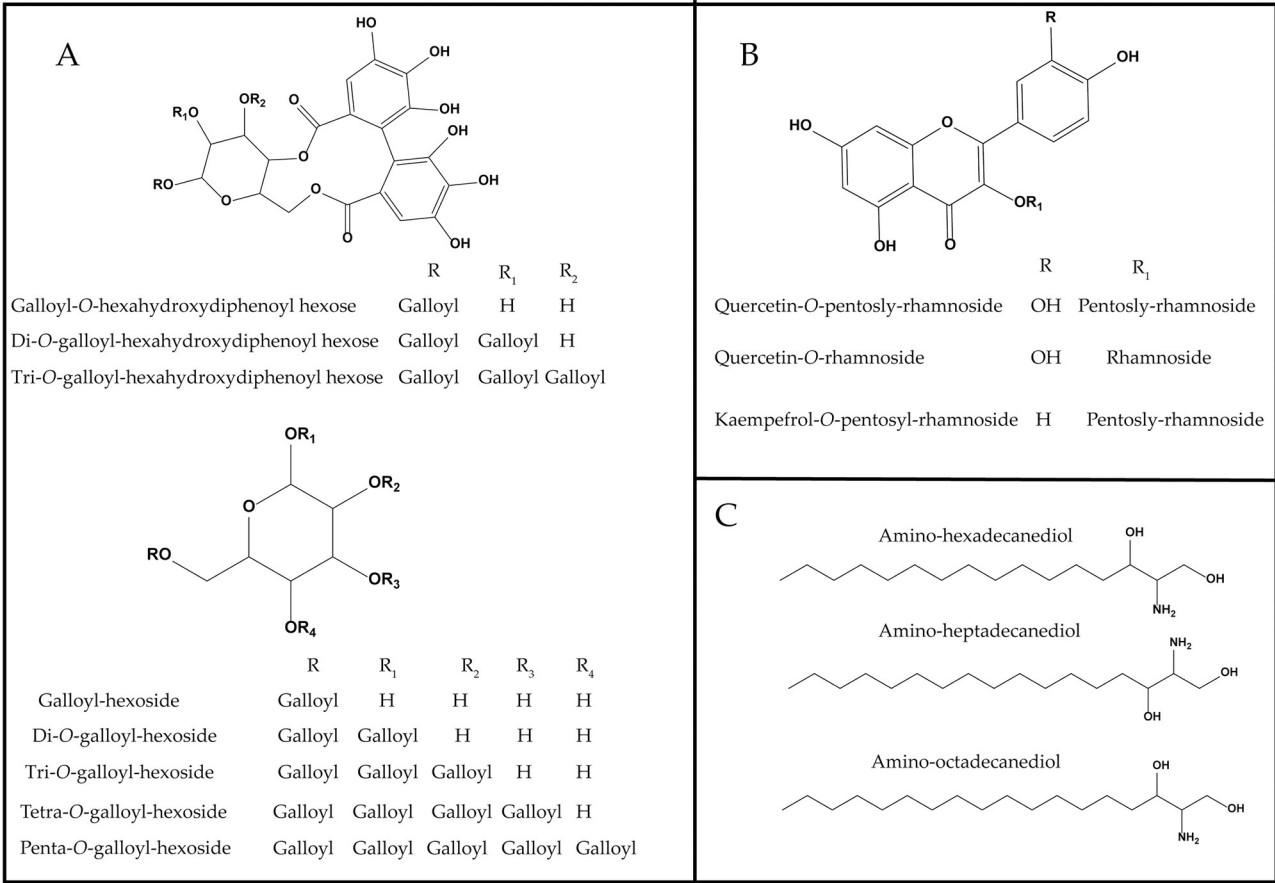

**Fig 2. Major classes of phytochemicals: (A) tannins, (B) flavonoids, (C) amino alcohols detected in *S. coriaceum* leaf crude extract.**

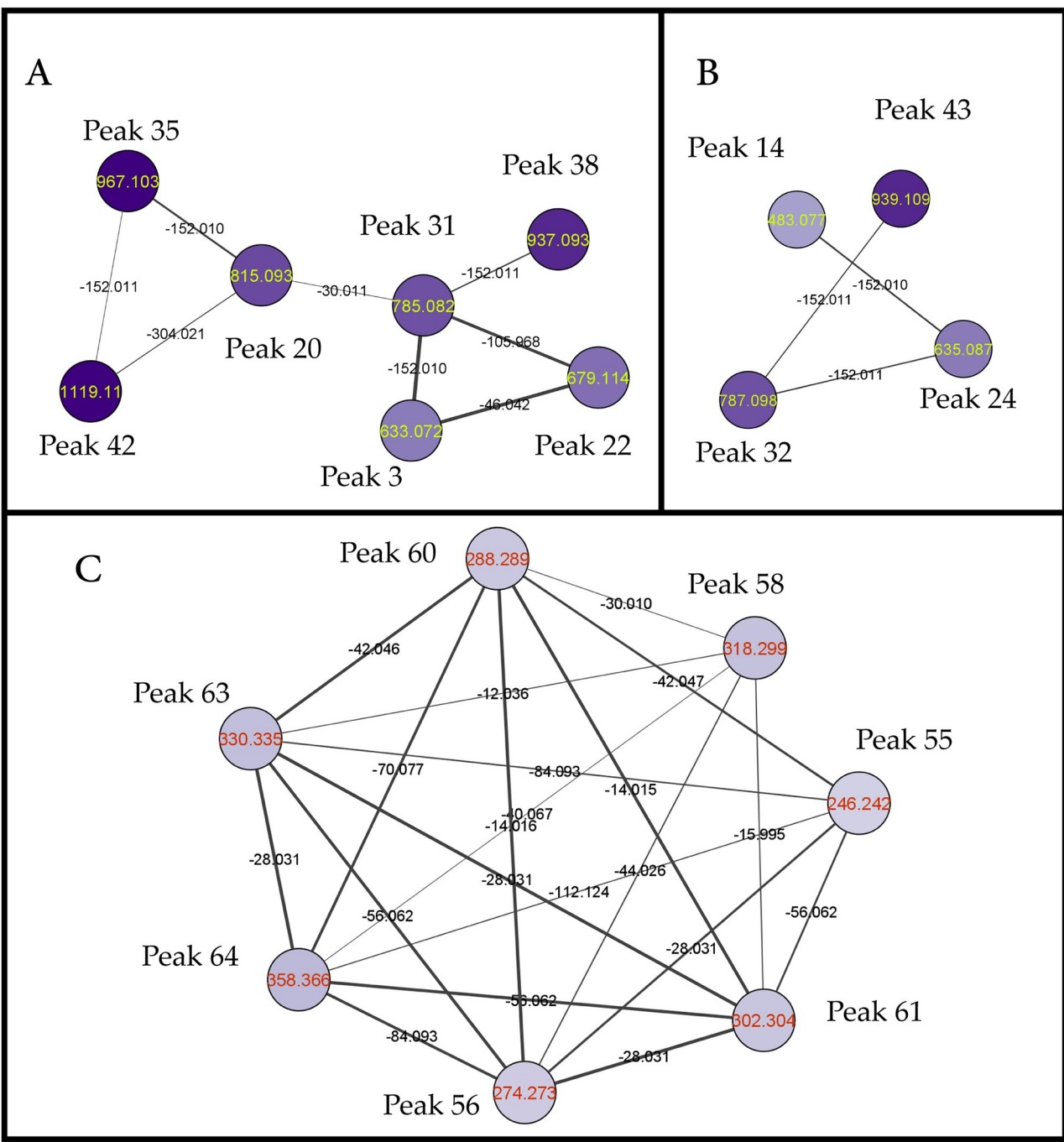

**Fig 3.** (A) Molecular network group of hexahydroxydiphenoyl hexose derived compounds. Peak numbers are according to that listed in (Table 1) as follows: peak 3; galloyl-*O*-hexahydroxydiphenoyl hexose, peak 20; methoxy-di-*O*-galloyl-hexahydroxydiphenoyl hexose, peak 22; ethoxygalloyl-*O*-hexahydroxydiphenoyl hexose, peak 31; di-*O*-galloyl-hexahydroxydiphenoyl hexose, peak 35; methoxy-tri-*O*-galloyl-hexahydroxydiphenoyl hexose, peak 38; tri-*O*-galloyl-hexahydroxydiphenoyl hexose, peak 42; methoxy-tetra-*O*-galloyl-hexahydroxydiphenoyl hexose. **(B)** Molecular network group of sugar polygalloyl esters. Peak numbers are according to that listed in (Table 1) as follows: peak 14; di-*O*-galloyl-hexoside, peak 24; tri-*O*-galloyl-hexoside, peak 32; tetra-*O*-galloyl-hexoside, peak 43; penta-*O*-galloyl-hexoside. **(C)** Molecular network group of amino alcohols measured under positive ionization mode. peak numbers are according to that listed in (Table 1) as follows: peak 55; amino-tetradecanediol, peak 56; amino-hexadecanediol, peak 58; amino-octadecanetriol, peak 60; amino-heptadecanediol, peak 61; amino-octadecanediol, peak 63; aminoeicosanediol, peak 64; aminodocosanediol. Nodes are color-coded in the three networks according to their *m/z* ranges and labelled by their parent ions. Edges are labelled by the mass differences between each node pair where their line widths are related to the cosine score (from thin, cosine score: 0.7, to thick, cosine score: 1).

that were either deoxyhexose or pentose with a respective loss of 146 and 132 amu [31]. In details, compounds 45 and 46 displayed [M-H] $^-$ at $m/z$ (579.1377, $C_{26}H_{27}O_{15}^-$) and $m/z$ (447.0941, $C_{21}H_{19}O_{11}^-$) annotated as quercetin-*O*-pentosyl-rhamnoside and quercetin-*O*-rhamnoside, respectively. Both compounds exhibited MS$^2$ spectrum (S3a and S3b Fig) of daughter ions at $m/z$ 300 indicative for quercetin radical anion and $m/z$ 271 following the loss of CO moiety typifying the fragmentation pattern reported for quercetin glycosides [32]. Other flavonol aglycones included kaempferol as detected in the peak 51 [M-H] $^-$ at $m/z$ (563.1411, $C_{26}H_{27}O_{14}^-$) and annotated as kaempferol-*O*-pentosyl-rhamnoside based on its MS$^2$ fragment ions (S3c Fig) at $m/z$ 284 ascribed to the radical anion of the kaempferol aglycone.

**Nitrogenous compounds and organic acids.**   Several unacylated long-chain amino alcohols, a class of sphingolipids, were detected in *S. coriaceum* crude extract *via* positive ionization mode as indicated from their even mass weights suggesting the presence of nitrogen atom. GNPS-aided molecular networking has successfully managed to group these closely related compounds (Fig 3C) such as amino-tetradecanediol (peak 55; 246.2425, $C_{14}H_{32}NO_2^+$), amino-hexadecanediol (peak 56; 274.2741, $C_{16}H_{36}NO_2^+$), amino-octadecanediol (peak 61; 302.3053, $C_{18}H_{40}NO_2^+$), aminoeicosanediol (peak 63; 330.3365, $C_{20}H_{44}NO_2^+$) and aminodocosanediol (peak 64; 358.3677, $C_{22}H_{48}NO_2^+$). Moreover, two amino triols peaks were also detected and annotated namely, amino-octadecanetriol (peak 58; 318.3003, $C_{18}H_{40}NO_3^+$) and aminoeicosanetriol (peak 62; 346.3309, $C_{20}H_{44}NO_3^+$) with a mass difference of 16 Da from their diols counterparts indicative for the extra hydroxyl group.

Other detected nitrogenous compounds in *S. coriaceum* extract included acylated amino acids derivatives annotated as *N*-deoxy-fructosyl tyrosine (peak 10; 182.0811, $C_9H_{12}NO_3^+$) and fructosyl-threonine (peak 10; 182.0811, $C_9H_{12}NO_3^+$) in addition to free amino acids i.e., tyrosine (peak 10; 182.0811, $C_9H_{12}NO_3^+$) and tryptophan (peak 17; 166.0862, $C_9H_{12}NO_2^+$). Both compounds exhibited a characteristic fragment at $m/z$ 91 indicative of the formation of the tropylium ion. Finally, few organic acids were also identified in this study including quinic acid (peak 2; 191.0572, $C_7H_{11}O_6^-$), shikimic acid (peak 6; 173.0456, $C_7H_9O_5^-$), citric acid (peak 7; 191.0201, $C_6H_7O_7^-$) and homocitric acid (peak 15; 207.0499, $C_7H_{11}O_7^+$) and eluting earlier considering their polarity.

## Antiproliferative activity

Lactate dehydrogenase leakage in the culture medium is an indication of compromised cell membrane integrity leading to cell death. The amount of LDH released is usually directly proportional to the number of lysed cells [33]. The cytotoxic effect of *S. coriaceum* methanolic extract against HepG2 cells was thus evaluated by quantifying the amount of LDH leaked into the culture medium. The amount of LDH released in *S. coriaceum* treated HepG2 cells at 40 µg/mL was significantly ($p \leq 0.0001$) higher (27%) than that in DMSO treated negative control (Fig 4A). The degree of LDH leaked gradually increased in a dose-dependent manner, peaking up to 47% (compared to DMSO control) at a dose of 100 µg/mL. The current results corroborated the findings of our previous study, whereby *S. coriaceum* exhibited pronounced cytotoxic activity against HepG2 cells, with a reported $IC_{50}$ value of 24.2 ± 2.8 µg/mL in methyl thiazolyl diphenyl-tetrazolium bromide (MTT) assay [12]. It is noteworthy that the same study reported *S. coriaceum* extract to be 2.6 fold less toxic to non-malignant HOE cells as compared to HepG2 cells.

The ability to evade growth suppressing signals and sustaining unlimited proliferation is a characteristic hallmark feature of cancer cells [34]. The half-maximal growth inhibitory activity of *S. coriaceum*, against HepG2 cells, was previously reported as 24.2 ± 2.8 µg/mL [12]. In

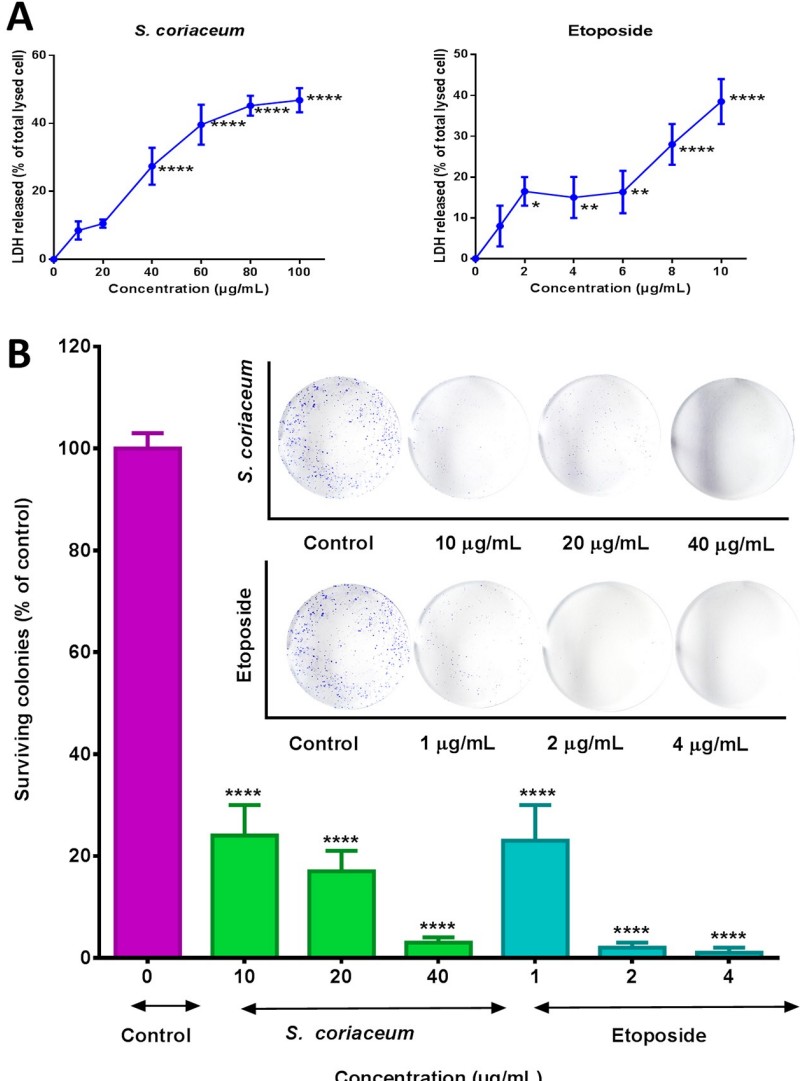

**Fig 4. Effect of *Syzygium coriaceum* leaf extract and etoposide on lactate dehydrogenase leakage (A), and clonogenic survival of HepG2 cells following 48 hours incubation.** Data are expressed as a percentage of total lysed cells for LDH and percentage surviving colonies compared to DMSO (0.125% v/v) for colony formation assay, respectively. The results are plotted as mean ± SEM of three independent experiments done in three replicates for each treatment. For colony formation assay, colony formed after 14 days were stained with 0.1% crystal violet and the images of the wells were captured using a digital camera. Asterisks represent significant differences between treated and untreated cells, * $p \leq 0.05$, ** $p \leq 0.01$, **** $p \leq 0.0001$.

accordance with the US National Cancer Institute guidelines, extracts with $IC_{50}$ values below 30 µg/mL are considered potent candidates for further investigations with regards to their anti-cancer potential [35]. Consequently, it is of value to gauge the *S. coriaceum* effect on the replicative ability of the cancer cells, at doses equivalent to ½ X, X and 2X $IC_{50}$ values. *S. coriaceum* was found to significantly inhibit the clonogenic capacity of HepG2, at all three test doses, with up to more than 90% at 40 µg/mL ($p \leq 0.0001$), in comparison to the negative control (Fig 4B). Likewise, etoposide caused a drastic reduction in the number of viable HepG2 colonies at the concentration range tested, with the effect being more pronounced at 2 µg/mL and

4 μg/mL. Moreover, etoposide at 4 μg/mL almost completely impeded the colony-forming ability of HepG2 cells. In addition, only 23 ± 7% of HepG2 cell populations survived treatment with 1 μg/mL of etoposide. A similar effect was achieved by 10 μg/mL of *S. coriaceum*, where only 24 ± 6% of HepG2 cells grew into distinct colonies. The 10 fold higher concentration of *S. coriaceum* needed to achieve a comparable activity (p > 0.05) to etoposide might be partly explained by the fact that the crude extract is comprised of a large pool of phytoconstituents, which might dilute the potency of the bioactive components. These results conclusively indicated that *S. coriaceum* exerted a long-term suppressive effect on HepG2 cells, by reducing the *in-vitro* cell's self-renewal ability in a dose-dependent manner. Although, low doses of the extract exhibited significant (p < 0.001) *in-vitro* antiproliferative activity against HepG2 cells, the beneficial effects of the extract can only be validated *in vivo* and human studies.

Flow cytometry analysis of HepG2 cells treated with 40 μg/mL *S. coriaceum* revealed a significant (p < 0.05) increase in both Annexin V-FITC and PI fluorescence level, compared to untreated control. The proportion of cells undergoing apoptotic/necrotic cell death, in HepG2 culture treated with 1 μg/mL etoposide was remarkably higher (p < 0.05) compared to cells treated with 10 μg/mL of *S. coriaceum* extract (Fig 5A & S4 Fig). Etoposide is known to induce apoptotic cell death in HepG2 cell as evidenced by the increased PARP and caspase 3 cleavages [36]. These results indicated that *S. coriaceum* induced both apoptotic and necrotic cell death in HepG2 cells. A similar effect was reported where *Lepidium sativum* extract induced both apoptotic and necrotic cell death in human breast cancer cells [37]. It is to be noted that, the cell membrane integrity is also compromised, leaking LDH, as a result of a late stage of apoptosis [38]. Moreover, secondary necrosis is known to be a natural outcome of the complete apoptotic pathway, in the absence of phagocytosis by scavenger cells [39]. *In vitro* cultures lack scavenger cells and therefore are unable to clear their corpse at the end of successful apoptotic events, thus initiating the process of secondary necrosis [40]. In this line, the present flow cytometric data is limited to distinguish the precise mode of induced cell death. Thus, further proteomic studies are warranted to establish the cellular events leading to *S. coriaceum* induced cell cytotoxicity in HepG2 cells.

In a recent study, *S. coriaceum* extract was reported to cause cell death in HepG2 cells by upregulating the production of intracellular ROS level and a parallel decrease in the intrinsic antioxidant enzyme activities, notably glutathione peroxidase [12]. *S. coriaceum* is reported to downregulate the gene expressions of microtubule-associated protein 1 light chain 3, beclin, and telomerase in MDA-MB-231 cells, all culminating in an increased rate of cell death [5]. Moreover, the same study also demonstrated the ability of *S. coriaceum* to suppress the expression of Bcl2 and BIRC5 antiapoptotic genes in MDA-MB-231 cells, highlighting that the antiproliferative activity of the extract may be partly mediated *via* the induction of apoptosis in cancer cells. Along similar lines, other Mauritian endemic *Syzygium* species, notably *S. commersonnii*, *S. mauritianum* and *S. venosum* were reported to suppress the proliferation of MDA-MB-231 and MCF-7 breast cancer cells, in a dose dependent manner [41].

Treatment of HepG2 cells with 20 μg/mL and 40 μg/mL *S. coriaceum* led to the accumulation of about 62.43% (p ≤ 0.005) and 72.14% (p ≤ 0.001), respectively, as opposed to 66.46% (untreated control cells) of proliferative cells in G0/G1 phase. The interruption of the cancer cell cycle at the G0/G1 phase is reported to subsequently trigger apoptotic pathways. The flow cytometric analysis confirmed that the extract induced cell death by arresting HepG2 cells in G0/G1 phase while etoposide-induced a G2/M phase arrest (Fig 5B & S5 Fig).

Mitochondrial damage caused by the loss of MMP has been proven to increase the mitochondrial membrane permeability, leaking apoptogenic factors into the cytosol and ultimately provoking apoptotic [24]. The effect of the extracts on mitochondrial membrane permeabilization was assessed by monitoring the change in MMP using JC-1 dye. Bright red fluorescence

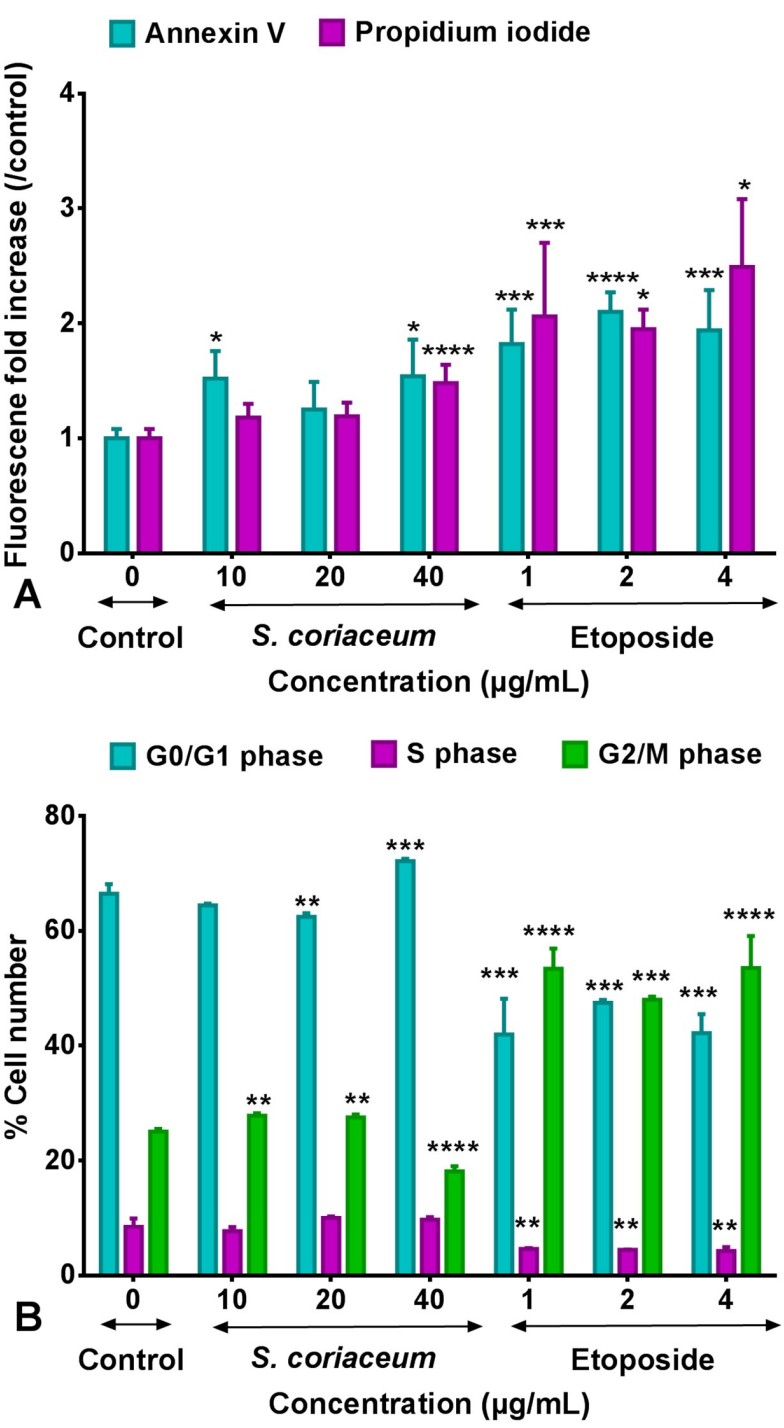

**Fig 5. Effect of *S. coriaceum* leaf extract and etoposide on HepG2 cells as analysed by flow cytometry.** (A) Annexin V-FITC/PI staining of HepG2 cells after 48 hours treatment and (B) Cell cycle progression. Apoptosis and necrosis levels are expressed as mean ± SD fold increase in Annexin V-FITC and PI fluorescence, respectively (n = 5). Percentage of cells in different phases (G0/G1, S and G2/M phases) are expressed as mean ± SD (n = 3). Asterisks represent significant differences between test concentrations and control. *p ≤ 0.05, **p ≤ 0.01,*** p ≤ 0.001, **** p ≤ 0.0001.

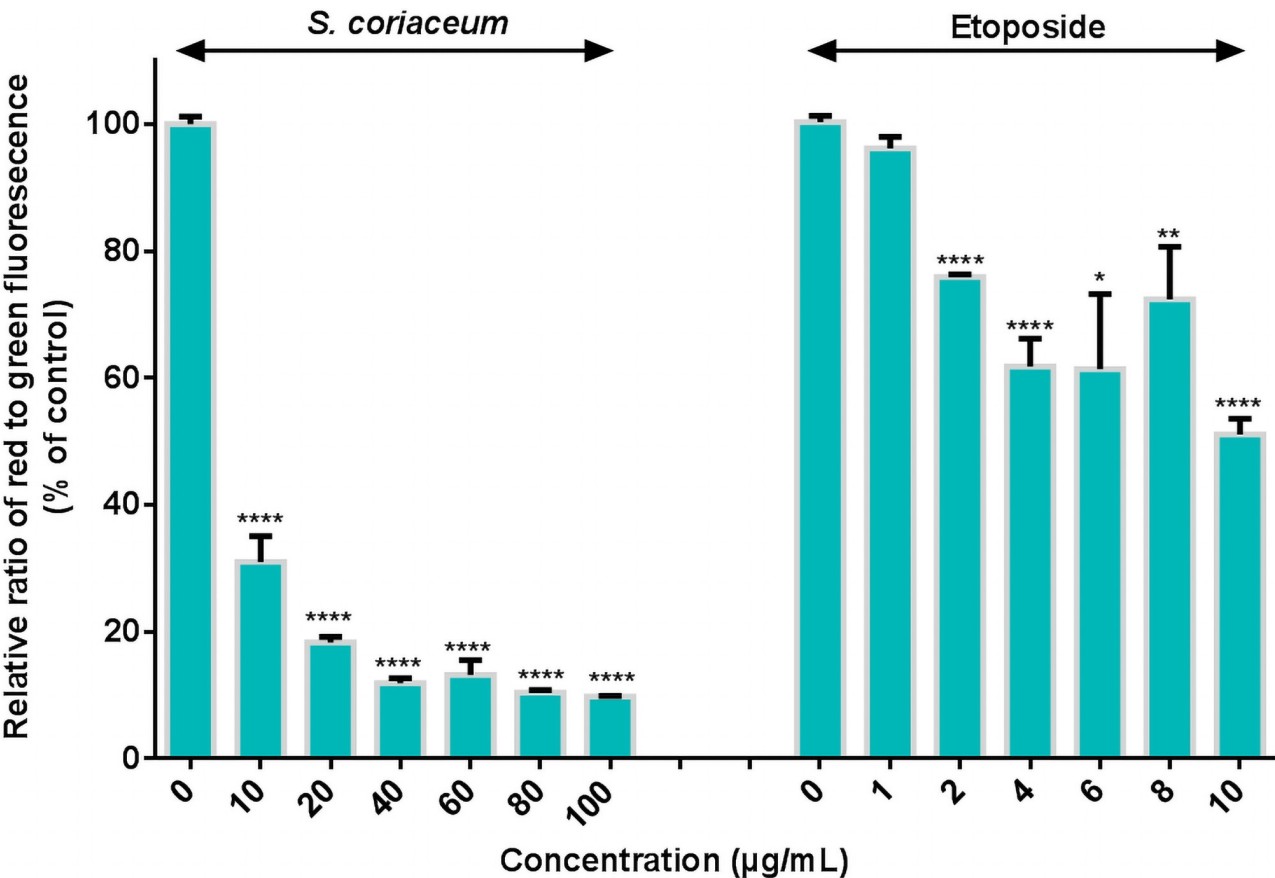

**Fig 6. Effect of *Syzygium coriaceum* leaf extract and etoposide on the mitochondrial membrane potential of HepG2 cells.** The ratio of red/green fluorescence units is given as a percentage of DMSO control. The results are plotted as mean ± SEM of three independent experiments done in three replicates for each treatment. Asterisks represent significant differences between treated and untreated cells, * $p \leq 0.05$, ** $p \leq 0.01$, **** $p \leq 0.0001$.

signalled healthy viable cells with high MMP, as opposed to cells with reduced MMP displaying green fluorescence [24]. Exposure to *S. coriaceum* extracts led to a significant ($p \leq 0.0001$) shift from red to green fluorescence signals in HepG2 cell culture, as indicated by the marked reduction in red/green fluorescence ratio of JC-1 dye compared to DMSO control (Fig 6) dropping to 9.8 ± 0.1% at 100 μg/mL as compared to untreated control. In similar experimental conditions, exposure of HepG2 cells to etoposide ranging between 2 μg/mL to 10 μg/mL, induced a dose dependent drop in red/green fluorescence ratio as opposed to control (Fig 6). These observations indicate that the antiproliferative activity of the extract against HepG2 cells, may in part be due to the collapse of mitochondrial membrane potential, further corroborating the probable induction of apoptotic mechanisms as revealed by flow cytometric analysis.

Plant extracts and/or the phytochemicals derived from them are known to provoke cancer cell death by causing oxidative DNA damage. For instance, celastrol was reported to induce DNA damage and subsequent cell cycle arrest, leading to a beneficial reduction in drug-resistant human colon cancer cells [42]. Hence, considering the G0/G1 phase arrest observed by the flow cytometric analysis in this study, it may be hypothesized that DNA damage may be a contributing mechanism through which *S. coriaceum* mediated its cytotoxic effect in HepG2 cells. In this view, *S. coriaceum* ability to cause DNA damage and subsequent induction of cell

**Table 2. Tail parameters of H₂O₂ and extract-treated HepG2 cells.**

| Extracts | Tail length | Tail intensity | Olive tail moment |
|---|---|---|---|
| Negative control | 32.8 ± 0.3 | 0.8 ± 0.1 | 0.2 ± 0.0 |
| *S. coriaceum* (10 µg/mL) | 53.2 ± 0.9**** | 16.7 ± 0.6**** | 3.8 ± 0.1**** |
| H₂O₂ (200 µM) | 90.8 ± 7.2**** | 46.8 ± 2.6**** | 14.1 ± 0.6**** |

Data are expressed as mean ± SEM of three independent experiments (n = 3). For each experiment, at least 100 comets were scored. Asterisks represent significant differences between treated and untreated cells,

****$p \leq 0.0001$.

cycle arrest in HepG2 cells were investigated using alkaline comet assay and flow cytometric analysis of propidium iodide stained cells. The degree of genetic damage induced by 10 µg/mL of the extract is reflected by the olive tail moment (Table 2 & S6 Fig), which is 19-fold increased ($p < 0.001$) compared to untreated control cells. The most potent DNA damage was observed in HepG2 cells treated with 200 µM H₂O₂ for 30 minutes, resulting in an olive tail moment 3.7 folds higher than *S. coriaceum* treated cells. The high olive tail moment of hydrogen peroxide may be explained by the fact that hydrogen peroxide is a common intracellular free-radical intermediate, generated in response to oxidative stress, and a known genotoxicant [43]. In contrast, *S. coriaceum* exhibited mild genotoxicity in HepG2 cells.

Many of the phytoconstituents identified using UPLC-MS from the *S.coriaceum* extract have been reported to exhibit similar anticancer mechanisms as reported by the crude extract. For example, di-*O*-galloyl-hexahydroxydiphenoyl hexose demonstrated both antiproliferative and multigene regulatory action in HepG2 cancer cells [44]. Ellagic acid impeded the growth of HepG2 by inducing G0/G1 phase arrest and ultimately triggering apoptotic cell death pathway [29]. Similarly, gallic acid and quercetin induced apoptosis by disrupting the cells' mitochondrial membrane potential, causing DNA damage and by arresting the cell cycle progression in G0/G1 phase in human leukaemia and HepG2 cells respectively [45]. The unacylated long-chain amino alcohols, sphingolipids and quinic acid present in *S.coriaceum* have been recognised as crucial regulators of pleiotropic cellular functions involved in cancer cell survival clinically [46,47].

## Conclusion

Overall, the present study demonstrated a comprehensive secondary metabolite profiling aiming towards providing a chemical basis for the short-term cytotoxic nature and the long-term clonal growth inhibitory activities of *S. coriaceum* against HepG2 cells. The current findings also provided evidence that *S. coriaceum* extracts triggered HepG2 cell death by inducing loss of mitochondrial membrane potential, causing cellular DNA damage with subsequently an arrest of the cell cycle progression in the G0/G1 phase. Non-targeted metabolites profiling *via* ultra-performance liquid chromatography coupled to high-resolution qTOF-MS enabled the separation as well as the annotation of 65 metabolites comprising tannins, flavonoids, nitrogenous compounds and organic acids, with tannins as the most abundant class in *S. coriaceum* methanolic leaf extract. Based on the literature, it may be concluded that both gallic acid and ellagic acid contributed, at least in part, to the extract's antiproliferative activity. Further research should be conducted to elucidate the contribution of the quercetin glycosides against the cytotoxicity of the extract in HepG2 cells, as well as to determine any synergistic action of the metabolites. Additionally, these promising results need to be further pursued in animal models or ideally in clinical studies to be more conclusive.

## Supporting information

**S1 Fig.** a. ESI-MS/MS spectrum of galloyl-*O*-hexahydroxydiphenoyl hexose (peak 3; $C_{27}H_{21}O_{18}^-$) in the negative ion mode. b. ESI-MS/MS spectrum of di-*O*-galloyl-hexahydroxy-diphenoyl hexose (peak 31; $C_{34}H_{25}O_{22}^-$) in the negative ion mode. c. ESI-MS/MS spectrum of tri-*O*-galloyl-hexahydroxydiphenoyl hexose (peak 38; $C_{41}H_{29}O_{26}^-$) in the negative ion mode. d. ESI-MS/MS spectrum of methoxy-di-*O*-galloyl-hexahydroxydiphenoyl hexose (peak 20; $C_{35}H_{27}O_{23}^-$) in the negative ion mode.
(TIF)

**S2 Fig.** a. ESI-MS/MS spectrum of galloyl-O-hexoside (peak 9; $C_{13}H_{15}O_{10}^-$) in the negative ion mode. b. ESI-MS/MS spectrum of di-*O*-galloyl rhamnoside (peak 29; $C_{20}H_{19}O_{13}^-$) in the negative ion mode.
(TIF)

**S3 Fig.** a. ESI-MS/MS spectrum of quercetin-*O*-pentosly-rhamnoside (peak 45; $C_{26}H_{27}O_{15}^-$) in the negative ion mode. b. ESI-MS/MS spectrum of quercetin-*O*-rhamnoside (peak 46; $C_{21}H_{19}O_{11}^-$) in the negative ion mode. c. ESI-MS/MS spectrum of kaempefrol-*O*-pentosyl-rhamnoside (peak 51; $C_{26}H_{27}O_{14}^-$) in the negative ion mode.
(TIF)

**S4 Fig. Representatives annexin V-FITC/PI flow cytometric profile of HepG2 cells, 48 hours post extract/control treatment.** Cells were treated for 48 hours with (A) negative control (0.025% DMSO); (B, C, D) 10 μg/ml, 20 μg/ml and 40 μg/ml of *Syzygium coriaceum*, respectively; (E, F, G) and 1 μg/ml, 2 μg/ml and 4 μg/ml of etoposide, respectively.
(TIF)

**S5 Fig. Representatives cell cycle histogram of HepG2 cells, 48 hours post extract/control treatment.** Cells were treated for 48 hours with (A) negative control (0.025% DMSO); (B, C, D) 10 μg/ml, 20 μg/ml and 40 μg/ml of *S. coriaceum*, respectively and (E, F, G) 1 μg/ml, 2 μg/ml and 4 μg/ml of etoposide, respectively.
(TIF)

**S6 Fig. Microscopic appearances of comet assay analysis.** DNA fragments in HepG2 cells after 24 hours exposure to (A) cell culture medium (negative control); (B) 200 μM $H_2O_2$ (positive control; 30 minutes) and (C) 10 μg/ml *Syzygium coriaceum* extract. Each figure represents a typical comet tail of the 100 observed cells from each experiment; magnification 200X, (n = 3).
(TIF)

## Acknowledgments

We thank the director and staff of Alteo Group, Médine Sugar Estate and Mauritius National Park Conservation Services under the Ministry of Agro-Industry & Food Security, Mauritius, for permission to collect endemic plant samples and the Mauritius Herbarium for plant identification.

## Author Contributions

**Conceptualization:** Vidushi S. Neergheen.

**Data curation:** Ahmed Serag.

**Formal analysis:** Nawraj Rummun, Ahmed Serag.

**Investigation:** Nawraj Rummun, Mohamed A. Farag.

**Methodology:** Srishti Ramsaha, Mohamed A. Farag, Vidushi S. Neergheen.

**Project administration:** Vidushi S. Neergheen.

**Resources:** Philippe Rondeau, Emmanuel Bourdon, Mohamed A. Farag.

**Software:** Philippe Rondeau, Emmanuel Bourdon.

**Supervision:** Theeshan Bahorun, Vidushi S. Neergheen.

**Validation:** Vidushi S. Neergheen.

**Writing – original draft:** Nawraj Rummun.

**Writing – review & editing:** Ahmed Serag, Philippe Rondeau, Srishti Ramsaha, Emmanuel Bourdon, Theeshan Bahorun, Mohamed A. Farag, Vidushi S. Neergheen.

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
