## [Decision Letter · Decision Letter 0]

10 Feb 2021

PONE-D-21-00692

Antiproliferative activity of Syzygium coriaceum, an endemic plant of Mauritius, with its UPLC-MS metabolite fingerprint: A mechanistic study

PLOS ONE

Dear Dr. Neergheen,

Thank you for submitting your manuscript to PLOS ONE. After careful consideration, we feel that it has merit but does not fully meet PLOS ONE’s publication criteria as it currently stands. Therefore, we invite you to submit a revised version of the manuscript that addresses the points raised during the review process.

The authors are advised to deepen both the Introduction and Discussion sections as proposed by the Reviewer #1. They should be notified they are not obliged whatsoever to cite the proposed articles. Elaboration of the text should be performed without any bias.

Additional efforts should be made towards quantification of the identified metabolites. Moreover, additional cell lines should be assessed. The mechanism of action of leaf extract should be proposed and supported by the experimental results.

Figures 2 A and 3 A are strangely stretched. Please revert to the original proportions. Resolution of Figure 1 and Figure 2 must be increased.

Additional clarifications of the methodology used are requested in the Reviewers' reports.

We look forward to receiving your revised manuscript.

Kind regards,

Branislav T. Šiler, Ph.D.

Academic Editor

PLOS ONE

Journal Requirements:

2. Please amend either the title on the online submission form (via Edit Submission) or the title in the manuscript so that they are identical.

3. Please include a caption for figure 5.

Reviewers' comments:

Reviewer's Responses to Questions

**Comments to the Author**

1. Is the manuscript technically sound, and do the data support the conclusions?

Reviewer #1: Yes

Reviewer #2: Partly

2. Has the statistical analysis been performed appropriately and rigorously? 

Reviewer #1: I Don't Know

Reviewer #2: N/A

3. Have the authors made all data underlying the findings in their manuscript fully available?

Reviewer #1: Yes

Reviewer #2: No

4. Is the manuscript presented in an intelligible fashion and written in standard English?

Reviewer #1: No

Reviewer #2: Yes

5. Review Comments to the Author

Reviewer #1: The authors tentatively profiled the phytoconstituents of the Syzygium coriaceum extract using UPLC-MS/MS and investigated its antiproliferative activities. There are some points to be improved.

1. The authors should improve their introduction and discussion by comparing the chemical composition and antiproliferative actives with other syzygium species. The following articles are recommended.

Sobeh, Mansour, Ahmed Esmat, Ganna Petruk, Mohamed AO Abdelfattah, Malak Dmirieh, Daria Maria Monti, Ashraf B. Abdel-Naim, and Michael Wink. "Phenolic compounds from Syzygium jambos (Myrtaceae) exhibit distinct antioxidant and hepatoprotective activities in vivo." Journal of Functional Foods 41 (2018): 223-231.

Sobeh, Mansour, Mona F. Mahmoud, Ganna Petruk, Samar Rezq, Mohamed L. Ashour, Fadia S. Youssef, Assem M. El-Shazly, Daria M. Monti, Ashraf B. Abdel-Naim, and Michael Wink. "Syzygium aqueum: A polyphenol-rich leaf extract exhibits antioxidant, hepatoprotective, pain-killing and anti-inflammatory activities in animal models." Frontiers in pharmacology 9 (2018): 566.

2. The authors should relatively quantify the compounds according to their relative abundance

3. The authors should improve the resolution of the figures.

4. The authors should investigate the antioxidant activities of the extract and quantify its phenolic content and then discuss the results.

5. The authors should use a professional software such as SigmaPlot or GraphPad to analyze the biological results

6. The antiproliferative effect and comet results should be extensively improved and discussed with previous reports. The following article is recommended.

Ghareeb MA, Mohamed T, Saad AM, Refahy LA, Sobeh M, Wink M. HPLC‐DAD‐ESI‐MS/MS analysis of fruits from Firmiana simplex (L.) and evaluation of their antioxidant and antigenotoxic properties. Journal of Pharmacy and Pharmacology. 2018 Jan;70(1):133-42.

Reviewer #2: The article is a good starting point for the evaluation of Syzygium coriaceum antiproliferative activity. It is well written and the characterization of the extract used in the treatment is very exhaustive. However, from my point of view, it is not suitable to publish in this journal in the present form. Major concerns are listed as following:

- What is the basis for the used S. coriaceum doses in the study? What is the translational relevance of these doses?

- The data shown is only in one cell lines. It is recommended to validate these results in more cancer cell lines. In addition, author should check or mention (if already known) the effect of S. coriaceum on normal (healthy) cells at the doses used in this study.

- Authors indicated potential mechanisms underlying S. coriaceum activity. However, the action pathway is still unclear. In that sense, it is recommended to evaluate the treatment effects on the gene expression and protein levels of the main targets by RT-PCR and Western blotting.

- In figure 4, in the nomenclature of the y-axis of the graphs, “vs % control” and “vs % of DMSO treated control” are used. Does it mean that, depending on the analysis, cells treated with DMSO (0.125% v/v) were used as control and in others, untreated cells?

- The authors also treat the cells with etoposide at various concentrations, but it is not always clear whether the comparison of results obtained with this compound with those obtained with S. coriaceum treatment is based on statistical analysis.

- Why do the authors generally refer to 40 ug/mL of S. coriaceum and 1 ug/mL of etoposide and not to the entire range of concentrations used for both?

- Why in some analyzes treatment with S. coriaceum is carried out for 24 h and in others for 48 h?

- At what concentration was the S. coriaceum extract used in the comet assay? (please specify it in the text and in the table). Why not to use more than one concentration (10, 20, 40 ug/ml)?

- Of the constituents identified in the S. coriaceum extract, which are considered to be the main responsible for the observed effects?

Other minor concerns are:

- In some cases the authors declare a % DMSO of 0.125% v/v and in others of 0.025% v/v. Please check.

- In the comet assay, how the concentration/time combination for H2O2 was selected?

- The results of Figure 5 are discussed earlier than those of Figure 4 (C, D). Please solve it, re-enumerating the figures or changing the order of the discussion.

6. PLOS authors have the option to publish the peer review history of their article (what does this mean?). If published, this will include your full peer review and any attached files.

Reviewer #1: No

Reviewer #2: No

---

## [Author Response · Author response to Decision Letter 0]

28 Mar 2021

The point-to-point response to the reviewer’s comments is provided below.

The reviewer`s comments are given in italic font and the authors response are given in bold font and highlighted yellow, for in-text changes. The format has been disrupted here. Please refer to the document uploaded.

Reviewer #1: The authors tentatively profiled the phytoconstituents of the Syzygium coriaceum extract using UPLC-MS/MS and investigated its antiproliferative activities. There are some points to be improved. 

1. The authors should improve their introduction and discussion by comparing the chemical composition and antiproliferative actives with other syzygium species. The following articles are recommended. 

Sobeh, Mansour, Ahmed Esmat, Ganna Petruk, Mohamed AO Abdelfattah, Malak Dmirieh, Daria Maria Monti, Ashraf B. Abdel-Naim, and Michael Wink. "Phenolic compounds from Syzygium jambos (Myrtaceae) exhibit distinct antioxidant and hepatoprotective activities in vivo." Journal of Functional Foods 41 (2018): 223-231. 

Sobeh, Mansour, Mona F. Mahmoud, Ganna Petruk, Samar Rezq, Mohamed L. Ashour, Fadia S. Youssef, Assem M. El-Shazly, Daria M. Monti, Ashraf B. Abdel-Naim, and Michael Wink. "Syzygium aqueum: A polyphenol-rich leaf extract exhibits antioxidant, hepatoprotective, pain-killing and anti-inflammatory activities in animal models." Frontiers in pharmacology 9 (2018): 566.

The introduction has been revised and the following insertion on page 4 line 89 to page 5 line 124 were made, that reads as follows:

S. coriaceum belongs to the Myrtaceae family. Syzygium Gaertn., which is the largest flowering genus within the family, encompasses about 1200 globally distributed species with 16 species native to Mauritius [6]. Plants from the genus Syzygium have particularly a long ethnomedicinal history worldwide with several reported biological activities [7,8]. Extracts from S. fruticosum and S. cumini exhibited anti-breast cancer activity in rat models, while S. aromaticum showed anti-colon cancer activity [8]. Leaf extract of S. aqueum has shown analgesic and antinociceptive potential in male Sprague-Dawley rats [9]. 

Investigation of Syzygium species native to Mauritius island revealed leaf extracts from S. commersonii, S. venosum, S. mauritianum and S. glomeratum to exhibit potent in-vitro antioxidant activities in terms of their reducing potential and free radical scavenging activity [10,11]. Likewise, the potent in-vitro activity of S. coriaceum leaf extract has been reported in multiple antioxidant assay models [5]. These findings further corroborated in a recent comparative study, whereby S. coriaceum was significantly more potent, compared to S. bijouxii and S. pyneei, in an array of six in-vitro antioxidant assay models [12]. Moreover, S. coriaceum leaf extract has been reported to significantly potentiate the transcriptional activities of antioxidant enzymes, notably glutathione peroxidase, in cultured COS7 cells [13]. 

Besides the aforementioned ability of S. coriaceum to mitigate oxidative stress, the pluripharmacological potential of the plant is also reported. The leaf extract potentiated the antibacterial activity of ampicillin against Staphylococcus aureus, Eschterichia coli and Pseudomonas aerigunosa [5]. Furthermore, essential oils isolated from S. coriaceum leaf, demonstrated potent anti α-amylase and anti tyrosinase enzyme activities [14]. The antiproliferative potential of S. coriaceum on breast cancer cells revealed that the methanolic leaf extract of S. coriaceum decreased the microtubule-associated protein 1 light chain 3, beclin and telomerase gene expressions, as well as inhibited the antiapoptotic gene expression. The authors attributed the effects to the presence of gallic acid and quercetin [5]. Recently, a preliminary cytotoxic screening of the S. coriaceum crude extract and fractions against liposarcoma cells (SW872), lung cancer cells (A549), liver cancer cells (HepG2) and non-malignant human ovarian cells (HOE) showed the extract selectivity against liver cancer cells [12]. The cytotoxic nature of S. coriaceum was attributed to its ability to upregulate intracellular oxidative stress level beyond a critical threshold, which was evident by a dose dependent surge in intracellular ROS level and a parallel decrease in glutathione peroxidase enzyme activity [12]. 

The phytochemistry of S. coriaceum leaf has been reported principally in terms of polyphenolic composition. Two major bioactive components contributing to the cytotoxic activity against HepG2 cells was elucidated as galllic acid and methyl gallate [12]. Several other secondary metabolites have been identified from S. coriaceum leaf, notably, gallotannins, quercetin glycosides, kaempferol glycosides, (+)- catechin, (-)- epicatechin gallate, procyanidin B1 dimer, (E)-β- ocimene and α-guaiene amongst others [12–14]. 

In line with the above background, this work thus aims to assess the anticancer effects of S. coriaceum and its mechanism of action against HepG2 cells related to its bioactive composition. High-resolution UPLC-MS aided by molecular networking has been employed to ultimately characterise the bioactive components, which could provide a chemical basis of the extract potential anticancer effects. In this context, molecular networking has been implemented, as it is a very efficient tool for the graphical investigation of structurally related metabolites or compound families thus enabling rapid identification of several phytochemicals within complex samples.

2. The authors should relatively quantify the compounds according to their relative abundance

Table 1 on page 15 is revised to include an additional column detailing the relative concentrations of the identified compounds. 

3. The authors should improve the resolution of the figures.

Figures with improved resolution are now included.

4. The authors should investigate the antioxidant activities of the extract and quantify its phenolic content and then discuss the results.

The antioxidant activities of the extracts has already been reported by our group, in a recent publication in the South African Journal of botany, 2020, volume 137, pp 1-10, entitled “Methyl gallate – Rich fraction of Syzygium coriaceum leaf extract induced cancer cell cytotoxicity via oxidative stress”. The reported antioxidant activities of the extract has been included in the revised introduction on page 4, line 92-101 which reads as follows, and the results discussed in line with the phytochemical constituents detected.

 Investigation of Syzygium species native to Mauritius island revealed leaf extracts from S. commersonii, S. venosum, S. mauritianum and S. glomeratum to exhibit potent in-vitro antioxidant activities in terms of their reducing potential and free radical scavenging activity [10,11]. Likewise, the potent in-vitro activity of S. coriaceum leaf extract has been reported in multiple antioxidant assay models [5]. These findings further corroborated in a recent comparative study, whereby S. coriaceum was significantly more potent, compared to S. bijouxii and S. pyneei, in an array of six in-vitro antioxidant assay models [12]. Moreover, S. coriaceum leaf extract has been reported to significantly potentiate the transcriptional activities of antioxidant enzyme, notably glutathione peroxidase, in cultured COS7 cells [13]. 

5. The authors should use a professional software such as SigmaPlot or GraphPad to analyze the biological results

GraphPad prism 6 software was used for all analyses of the biological results. This was mentioned in the Statistical analysis section on page 12 line 279-283, which read as follows: 

Statistical analyses were performed using GraphPad Prism 6 software (GraphPad Inc., San Diego, California). The mean values among extract and control were compared using One-Way ANOVA. Student t-test and/or Tukey’s multiple comparisons as Post Hoc test was used to assess significant differences between the mean value for extract concentrations and that of the negative control.

6. The antiproliferative effect and comet results should be extensively improved and discussed with previous reports. The following article is recommended. 

Ghareeb MA, Mohamed T, Saad AM, Refahy LA, Sobeh M, Wink M. HPLC‐DAD‐ESI‐MS/MS analysis of fruits from Firmiana simplex (L.) and evaluation of their antioxidant and antigenotoxic properties. Journal of Pharmacy and Pharmacology. 2018 Jan;70(1):133-42.

The discussion pertaining to the antiproliferative effect of the extract has been revised on page 30- 32, lines 534- 554 is revised to read as follows:

It is to be noted that, the cell membrane integrity is also compromised, leaking LDH, as a result of late stage of apoptosis [38]. Moreover, secondary necrosis is known to be a natural outcome of the complete apoptotic pathway, in the absence of phagocytosis by scavenger cells [39]. In-vitro cultures lack scavenger cells and therefore are unable to clear their corpse at the end of successful apoptotic events, thus initiating the process of secondary necrosis [40]. In this line the present flow cytometric data is limited to distinguish the precise mode of induced cell death. Thus, further proteomic studies is warranted to establish the cellular events leading to S. coriaceum induced cell cytotoxicity in HepG2 cells. 

 In a recent study, S. coriaceum extract was reported to cause cell death in HepG2 cells by upregulating of the production of intracellular ROS level and a parallel decrease in the intrinsic antioxidant enzyme activities, notably glutathione peroxidase [12]. S. coriaceum is reported to downregulate the gene expressions of microtubule-associated protein 1 light chain 3, beclin, and telomerase in MDA-MB-231 cells, all culminating in an increased rate of cell death [5]. Moreover, the same study also demonstrated the ability of S. coriaceum to suppress the expression of Bcl2 and BIRC5 antiapoptotic genes in MDA-MB-231 cells, highlighting that the antiproliferative activity of the extract may be partly mediated via the induction of apoptosis in cancer cells. Along similar lines, other Mauritian endemic Syzygium species, notably S. commersonnii, S. mauritianum and S. venosum were reported to suppress the proliferation of MDA-MB-231 and MCF-7 breast cancer cells, in a dose dependent manner [41]. 

Reviewer #2: The article is a good starting point for the evaluation of Syzygium coriaceum antiproliferative activity. It is well written and the characterization of the extract used in the treatment is very exhaustive. However, from my point of view, it is not suitable to publish in this journal in the present form. Major concerns are listed as following:

- What is the basis for the used S. coriaceum doses in the study? What is the translational relevance of these doses?

The dose selected for this study was based on the findings of our previous report published in the South African journal of botany, 2020, volume 137, pp 1-10 entitled, “Methyl gallate – Rich fraction of Syzygium coriaceum leaf extract induced cancer cell cytotoxicity via oxidative stress”; as well as based on the criteria of the US National Cancer Institute for crude extracts. It is worth mentioning that the current work is an exploratory study and thus we used doses equivalent to half IC50, IC50 and 2X IC50 values for some assays. Although, the doses used are minimal, their beneficial effects can only be validated in-vivo and in-human studies to establish their translational relevance. The relevant methodology section on page 9 line 220-221 and page 10 line 234 - 236 as well as the discussion on pages 30-31 lines 486-524, has been revised to include the rational for the doses used in this study. 

Page 9 line 220-221: 

Following the treatment period of 48 hours with different concentrations of extract (10, 20 and 40 µg/mL, equivalent to ½ X , X and 2X of the reported IC50 value) and control, the media was replaced with fresh complete culture media and cells were grown under standard recommended culture conditions for 7 additional days.

Page 10 line 234 – 236: 

HepG2 cells (3 x 104 cells/well) were cultured in 12-well plates overnight. Following trial experiements using different dilutions of extract from 10 µg/mL to 40 µg/mL at 24 hours and 48 hours, the cells were exposed to 10 µg/mL extract for 24 hours. The negative control consisted of untreated HepG2 cells while the positive control cells were treated with 200 µM H2O2 for 30 minutes as previously reported [21]. Cells were washed with PBS, harvested using 1 X trypsin- EDTA and re-suspended in 100 µL of PBS. 

Pages 30-31 lines 486-524:

The ability to evade growth suppressing signals and sustaining unlimited proliferation is a characteristic hallmark feature of cancer cells [34]. The half maximal growth inhibitory activity of S. coriaceum, against HepG2 cells, was previously reported as 24.2 ± 2.8 µg/mL [12]. In accordance to the US National Cancer Institute guidelines, extracts with IC50 values below 30 µg/mL are considered potent candidates for further investigations with regards to their anticancer potential [35]. Consequently, it is of value to gauge S. coriaceum effect on the replicative ability of the cancer cells, at doses equivalent to ½ X, X and 2X IC50 values. S. coriaceum was found to significantly inhibit the clonogenic capacity of HepG2, at all three test doses, with up to more than 90 % at 40 µg/mL (p ≤ 0.0001), in comparison to the negative control (Fig 4 B). Likewise, etoposide caused a drastic reduction in the number of viable HepG2 colonies at the concentration range tested, with the effect being more pronounced at 2 µg/mL and 4 µg/mL. Moreover, etoposide at 4 µg/mL almost completely impeded the colony-forming ability of HepG2 cells. In addition, only 23 ± 7 % of HepG2 cell populations survived treatment with 1 µg/mL of etoposide. A similar effect was achieved by 10 µg/mL of S. coriaceum, where only 24 ± 6 % of HepG2 cells grew into distinct colonies. The 10 folds higher concentration of S. coriaceum needed to achieve a comparable activity (p > 0.05) to etoposide might be partly explained by the fact that the crude extract is comprised of a large pool of phytoconstituents, which might dilute the potency of the bioactive components. These results conclusively indicated that S. coriaceum exerted a long-term suppressive effect on HepG2 cells, by reducing the in-vitro cell`s self-renewal ability in a dose-dependent manner. Although, low doses of the extract exhibited significant (p < 0.001) in-vitro antiproliferative activity against HepG2 cells, the beneficial effects of the extract can only be validated in-vivo and in human studies. 

- The data shown is only in one cell lines. It is recommended to validate these results in more cancer cell lines. In addition, author should check or mention (if already known) the effect of S. coriaceum on normal (healthy) cells at the doses used in this study.

This study is a continuation of a recently published investigation in the South African journal of botany, 2020, volume 137, pp 1-10 entitled, “Methyl gallate – Rich fraction of Syzygium coriaceum leaf extract induced cancer cell cytotoxicity via oxidative stress”, whereby the extract cytotoxicity was compared in different cell lines and the growth of HepG2 cells were highly suppressed. The reported effect of the extract on the different cancer cell lines are included in the introduction on page 5, lines 116-121; while the effect on healthy cells are included in the discussion on page 42 lines 423-424.

Page 5, lines 116-121:

Recently, a preliminary cytotoxic screening of the S. coriaceum crude extract and fractions against liposarcoma cells (SW872), lung cancer cells (A549), liver cancer cells (HepG2) and non-malignant human ovarian cells (HOE) showed the extract selectivity against liver cancer cells [12]. The cytotoxic nature of S. coriaceum was attributed its ability to upregulate intracellular oxidative stress level beyond a critical threshold, which was evident by a dose dependent surge in intracellular ROS level and a parallel decrease in glutathione peroxidase enzyme activity [12].

Page 29 lines 470-472:

It is noteworthy that the same study reported S. coriaceum extract to be 2.6 fold less toxic to non-malignant HOE cells as compared to HepG2 cells.

- Authors indicated potential mechanisms underlying S. coriaceum activity. However, the action pathway is still unclear. In that sense, it is recommended to evaluate the treatment effects on the gene expression and protein levels of the main targets by RT-PCR and Western blotting.

We take good note of the reviewers comment. The objective of the current study was focused on the metabolic profiling of the extract in relation to its cytotoxic effect. The recommendation to evaluate the influence of the extract on the cell proteomic`s is being envisaged in a future study. The discussion on page 31-32, lines 541- 546 is revised to include the following: 

In this line the present flow cytometric data is limited to distinguish the precise mode of induced cell death. Thus, further proteomic studies are warranted to establish the cellular events leading to S. coriaceum induced cell cytotoxicity in HepG2 cells.

- In figure 4, in the nomenclature of the y-axis of the graphs, “vs % control” and “vs % of DMSO treated control” are used. Does it mean that, depending on the analysis, cells treated with DMSO (0.125% v/v) were used as control and in others, untreated cells?

The negative control comprised of 0.125 % v/v DMSO. This has been included in the methodology section on page 9 lines 179-181. Also the nomenclature of the y-axis has been standardised as “% of control” in the revised figure 4. 

- The authors also treat the cells with etoposide at various concentrations, but it is not always clear whether the comparison of results obtained with this compound with those obtained with S. coriaceum treatment is based on statistical analysis.

The statistical “p” value is included accordingly on page 30 line 504 and on page 31 line 530.

- Why do the authors generally refer to 40 ug/mL of S. coriaceum and 1 ug/mL of etoposide and not to the entire range of concentrations used for both?

The comparison of the entire range of concentration is now included in the discussion section which now reads as follows;

Page 30, lines 495-499:

S. coriaceum was found to significantly inhibit the clonogenic capacity of HepG2, at all three test doses, with up to more than 90 % at 40 µg/mL (p ≤ 0.0001), in comparison to the negative control (Fig 4 B). Likewise, etoposide caused a drastic reduction in the number of viable HepG2 colonies at the concentration range tested, with the effect being more pronounced at 2 µg/mL and 4 µg/mL.

Page 32, lines 560-562:

Treatment of HepG2 cells with 20 µg/mL and 40 µg/mL S. coriaceum led to the accumulation of about 62.43 % (p ≤ 0.005) and 72.14 % (p ≤ 0.001), respectively, as opposed to 66.46 % (untreated control cells) of proliferative cells in G0/G1 phase.

Page 33, lines 576-582:

Exposure to S. coriaceum extracts led to a significant (p ≤ 0.0001) shift from red to green fluorescence signals in HepG2 cell culture, as indicated by the marked reduction in red/green fluorescence ratio of JC-1 dye compared to DMSO control (Fig 6) dropping to 9.8 ± 0.1 % at 100 µg/mL as compared to untreated control. In similar experimental conditions, exposure of HepG2 cells to etoposide ranging between 2 µg/ml to 10 µg/ml, induced a dose dependent drop in red/green fluorescence ratio as opposed to control (Fig 6).

- Why in some analyzes treatment with S. coriaceum is carried out for 24 h and in others for 48 h?

- At what concentration was the S. coriaceum extract used in the comet assay? (please specify it in the text and in the table). Why not to use more than one concentration (10, 20, 40 ug/ml)?

All experiments, except the comet assay, were carried out for 48 hours. The rational for carrying out the comet assay for 24 hours is now included in the methodology section on page 10, lines 234-236, which read as follows:

Following trial experiments using different dilutions of extract from 10 µg/mL to 40 µg/mL at 24 hours and 48 hours, the cells were exposed to 10 µg/mL extract for 24 hours. 

The concentration of S. coriaceum used for comet assay is included in text on page 48 in table 2.

- Of the constituents identified in the S. coriaceum extract, which are considered to be the main responsible for the observed effects?

Both Gallic acid and Methyl gallate has been previously reported to be responsible for the extracts observed cytotoxicty in HepG2 cells. The discussion on pages 26-27, lines 403-414 has been revised to now read as: 

Methyl gallate was the most abundant metabolite quantified followed by gallic acid (Table 1). It is worth noting that high total phenolic content (62 ± 6.04 mg gallic acid equivallent per gram extract) of S. coriaceum leaf has been reported; with both gallic acid and methyl gallate identified as major bioactive compounds contributing to its cytotoxicity against HepG2 cells, via upregulation of intracellular reactive oxygen species [12]. Likewise, ellagic acid has been shown to suppress the growth of HepG2 cells via upregulation of intracellular ROS level, increased capspase- 3/9 activity as well as increased expression of P53, Bax and Cyt-c apoptosis-related proteins, subsequently inducing G0/G1 cell cycle arrest and ultimately apoptotic cell death [29]. However, the contribution of other identified constituents with %RSD above 20 (Table 1), notably, quinic acid, O-galloylyglycerol, Di-O-gallolyl-hexoside and quercetin-O-rhamnoside cannot be neglected. 

Other minor concerns are: 

- In some cases the authors declare a % DMSO of 0.125% v/v and in others of 0.025% v/v. Please check.

This is correct. The extract was dissolved in a vehicle comprising of 0.025 % v/v for the flow cytometric analysis only. 

- In the comet assay, how the concentration/time combination for H2O2 was selected?

This concentration/time combination was selected based on a previously reported protocol. This has been inserted in the included in the methodology section, on page 10, lines 236-238, as follows:

The negative control consisted of untreated HepG2 cells while the positive control cells were treated with 200 µM H2O2 for 30 minutes as previously reported [21].

- The results of Figure 5 are discussed earlier than those of Figure 4 (C, D). Please solve it, re-enumerating the figures or changing the order of the discussion.

Figure 4 (C,D) has been re-numerbered as Figure 6.

---

## [Decision Letter · Decision Letter 1]

13 Apr 2021

PONE-D-21-00692R1

Antiproliferative activity of Syzygium coriaceum, an endemic plant of Mauritius, with its UPLC-MS metabolite fingerprint: A mechanistic study

PLOS ONE

Dear Dr. Neergheen,

Thank you for submitting your manuscript to PLOS ONE. After careful consideration, we feel that it has merit but does not fully meet PLOS ONE’s publication criteria as it currently stands. Therefore, we invite you to submit a revised version of the manuscript that addresses the points raised during the review process.

The text still contains typographic errors which have to be resolved before accepting for publication. Moreover, language usage should be further polished in orded to improve general readability of the manuscript. Please do not use random word capitalization in both the body text and subtitles.

L172: Please define "100 U/L streptomycin-penicillin".

L225: "Flow cytometry analysis" should stand instead.

L250: Metabolite fingerprint

Table 1: "Hexahydroxydiphenoylproto-quercitoL", "L" stands in upper case.

L319-335: Please do not capitalize random words or compound names unless they stand at the beginning of a sentence.

We look forward to receiving your revised manuscript.

Kind regards,

Branislav T. Šiler, Ph.D.

Academic Editor

PLOS ONE

Journal Requirements:

Reviewers' comments:

Reviewer's Responses to Questions

**Comments to the Author**

1. If the authors have adequately addressed your comments raised in a previous round of review and you feel that this manuscript is now acceptable for publication, you may indicate that here to bypass the “Comments to the Author” section, enter your conflict of interest statement in the “Confidential to Editor” section, and submit your "Accept" recommendation.

Reviewer #1: All comments have been addressed

2. Is the manuscript technically sound, and do the data support the conclusions?

Reviewer #1: Yes

3. Has the statistical analysis been performed appropriately and rigorously? 

Reviewer #1: Yes

4. Have the authors made all data underlying the findings in their manuscript fully available?

Reviewer #1: Yes

5. Is the manuscript presented in an intelligible fashion and written in standard English?

Reviewer #1: Yes

6. Review Comments to the Author

Reviewer #1: (No Response)

7. PLOS authors have the option to publish the peer review history of their article (what does this mean?). If published, this will include your full peer review and any attached files.

Reviewer #1: No

---

## [Author Response · Author response to Decision Letter 1]

12 May 2021

The point-to-point response to the reviewer’s comments is provided below.

1. The text still contains typographic errors which have to be resolved before accepting for publication. Moreover, language usage should be further polished in orded to improve general readability of the manuscript. Please do not use random word capitalization in both the body text and subtitles.

The typographic errors have been resolved and the readability of the manuscript has been checked. Random word capitalization has been deleted both in text and in subtitles.

2. L172: Please define "100 U/L streptomycin-penicillin".

Line 176 now reads as “100 units /L penicillin and 100 units /L streptomycin”.

3. L225: "Flow cytometry analysis" should stand instead.

Line 231 now reads as “Flow cytometry analysis”

4. L250: Metabolite fingerprint

Line 231 now reads as “Metabolite fingerprint of S. coriaceum leaf extract”

5. Table 1: "Hexahydroxydiphenoylproto-quercitoL", "L" stands in upper case.

"Hexahydroxydiphenoylproto-quercitoL" in table 1 now reads as “Hexahydroxydiphenoylproto-quercitol”.

6. L319-335: Please do not capitalize random words or compound names unless they stand at the beginning of a sentence.

Lines 324 to 347 are revised to now read as follows:

“Fig 3: (A) Molecular network group of hexahydroxydiphenoyl hexose derived compounds. Peak numbers are according to that listed in (Table 1) as follows: peak 3; galloyl-O-hexahydroxydiphenoyl hexose, peak 20; methoxy-di-O-galloyl-hexahydroxydiphenoyl hexose, peak 22; ethoxygalloyl-O-hexahydroxydiphenoyl hexose, peak 31; di-O-galloyl-hexahydroxydiphenoyl hexose, peak 35; methoxy-tri-O-galloyl-hexahydroxydiphenoyl hexose, peak 38; tri-O-galloyl-hexahydroxydiphenoyl hexose, peak 42; methoxy-tetra-O-galloyl-hexahydroxydiphenoyl hexose. (B) Molecular network group of sugar polygalloyl esters. Peak numbers are according to that listed in (Table 1) as follows: peak 14; di-O-galloyl-hexoside, peak 24; tri-O-galloyl-hexoside, peak 32; tetra-O-galloyl-hexoside, peak 43; penta-O-galloyl-hexoside. (C) Molecular network group of amino alcohols measured under positive ionization mode. peak numbers are according to that listed in (Table 1) as follows: peak 55; amino-tetradecanediol, peak 56; amino-hexadecanediol, peak 58; amino-octadecanetriol, peak 60; amino-heptadecanediol, peak 61; amino-octadecanediol, peak 63; aminoeicosanediol, peak 64; aminodocosanediol. Nodes are color-coded in the three networks according to their m/z ranges and labelled by their parent ions. Edges are labelled by the mass differences between each node pair where their line widths are related to the cosine score (from thin, cosine score: 0.7, to thick, cosine score: 1).”

---

## [Editor Report · Decision Letter 2]

14 May 2021

Antiproliferative activity of Syzygium coriaceum, an endemic plant of Mauritius, with its UPLC-MS metabolite fingerprint: A mechanistic study

PONE-D-21-00692R2

Dear Dr. Neergheen,

We’re pleased to inform you that your manuscript has been judged scientifically suitable for publication and will be formally accepted for publication once it meets all outstanding technical requirements.

Kind regards,

Branislav T. Šiler, Ph.D.

Academic Editor

PLOS ONE
---

## [Editor Report · Acceptance letter]

21 May 2021

PONE-D-21-00692R2 

Antiproliferative activity of *Syzygium coriaceum*, an endemic plant of Mauritius, with its UPLC-MS metabolite fingerprint: A mechanistic study 

Dear Dr. Neergheen:

I'm pleased to inform you that your manuscript has been deemed suitable for publication in PLOS ONE. Congratulations! Your manuscript is now with our production department. 

Kind regards, 

on behalf of

Dr. Branislav T. Šiler 

Academic Editor

PLOS ONE